# RectifID: Personalizing Rectified Flow with Anchored Classifier Guidance

**Zhicheng Sun**[1] , **Zhenhao Yang**[3]**, Yang Jin**[1]**, Haozhe Chi**[1]**, Kun Xu**[2]**, Kun Xu**[2]**,**
**Liwei Chen**[2]**, Hao Jiang**[1]**, Yang Song, Kun Gai**[2]**, Yadong Mu**[1]*

[1]Peking University, [2]Kuaishou Technology,
[3]University of Electronic Science and Technology of China
{sunzc,myd}@pku.edu.cn

## Abstract

Customizing diffusion models to generate identity-preserving images from user-provided reference images is an intriguing new problem. The prevalent approaches typically require training on extensive domain-specific images to achieve identity preservation, which lacks flexibility across different use cases. To address this issue, we exploit classifier guidance, a training-free technique that steers diffusion models using an existing classifier, for personalized image generation. Our study shows that based on a recent rectified flow framework, the major limitation of vanilla classifier guidance in requiring a special classifier can be resolved with a simple fixed-point solution, allowing flexible personalization with off-the-shelf image discriminators. Moreover, its solving procedure proves to be stable when anchored to a reference flow trajectory, with a convergence guarantee. The derived method is implemented on rectified flow with different off-the-shelf image discriminators, delivering advantageous personalization results for human faces, live subjects, and certain objects. Code is available at `https://github.com/feifeiobama/RectifID`.

## 1 Introduction

Recent advances in diffusion models (Sohl-Dickstein et al., 2015; Song and Ermon, 2019; Ho et al., 2020) have ignited a surge of research into their customizability. A prominent example is personalized image generation, which aims to integrate user-defined subjects into the generated image. This plays a pivotal role in AI art creation, empowering users to produce identity-consistent images with greater customizability beyond text prompts. Nevertheless, there remain significant challenges in accurately preserving the subject's identity and being flexible to a variety of personalization needs.

Existing personalization methods are limited in these two aspects, as they require an extra finetuning or pre-training stage. For example, the pioneering works (Gal et al., 2023; Ruiz et al., 2023a) finetune conditional embeddings or model parameters per subject, resulting in suboptimal efficiency and identity consistency due to lack of domain knowledge. On the other hand, the recently prevailing tuning-free methods (Wei et al., 2023; Ye et al., 2023; Li et al., 2024; Wang et al., 2024b) pre-train a conditioning adapter to encode subject features into the generation process. However, their models must be pre-trained on extensive domain-specific data, *e.g.* LAION-Face 50M (Zheng et al., 2022), which is costly in the first place, and cannot be transferred flexibly across different data domains, *e.g.* from human faces to common live subjects and objects, and even to multiple subjects.

To address both challenges of identity consistency and flexibility, we advocate a training-free approach that utilizes the guidance of a pre-trained discriminator without extra training of the generative model. This methodology is well-known as classifier guidance (Dhariwal and Nichol, 2021), which modifies

---

*Corresponding author.

38th Conference on Neural Information Processing Systems (NeurIPS 2024).

Figure 1: Illustration of training-free classifier guidance. Left: an off-the-shelf discriminator can be reused to steer the existing diffusion model, *e.g.* rectified flow, to generate identity-preserving images. Right: personalized image generation results for human faces and objects using our proposed method.

an existing denoising process using the gradient from a pre-trained classifier. The rationale behind our exploitation is twofold: first, it directly harnesses the discriminator's domain knowledge for identity preservation, which may be a cost-effective substitute for training on domain-specific datasets; secondly, keeping the diffusion model intact allows for plug-and-play combination with different discriminators, as shown in Fig. 1, which enhances its flexibility across various personalization tasks. However, the original classifier guidance is largely limited in the reliance on a special classifier trained on noised inputs. Despite recent efforts to approximate the guidance (Kim et al., 2022a; Liu et al., 2023b; Wallace et al., 2023; Ben-Hamu et al., 2024), they have mainly focused on computational efficiency, and have yet to achieve sophisticated performance on personalization tasks.

Technically, to extend classifier guidance for personalized image generation, our work builds on a recent framework named rectified flow (Liu et al., 2023a) featuring strong theoretical properties, *e.g.* the straightness of its sampling trajectory. By approximating the rectified flow to be ideally straight, the original classifier guidance is reformulated as a simple fixed-point problem concerning only the trajectory endpoints, thus naturally overcoming its reliance on a special noise-aware classifier. This allows flexible reuse of image discriminators for identity preservation in personalization tasks. Furthermore, we propose to anchor the classifier-guided flow trajectory to a reference trajectory to improve the stability of its solving process, which provides a convergence guarantee in theoretical scenarios and proves even more crucial in practice. Lastly, a clear connection is established between our derived anchored classifier guidance and the existing approximation practices.

The derived method is implemented for a practical class of rectified flow (Yan et al., 2024) assumed to be piecewise straight, in combination with face or object discriminators. This provides flexibility for a range of personalization tasks on human faces, live subjects, certain objects, and multiple subjects. Extensive experimental results on these tasks clearly validate the effectiveness of our approach. Our contributions are summarized as follows: (1) We propose a training-free approach to flexibly personalize rectified flow, based on a fixed-point formulation of classifier guidance. (2) To improve its stability, we anchor the flow trajectory to a reference trajectory, which yields a theoretical convergence guarantee when the flow is ideally straight. (3) The proposed method is implemented on a relaxed piecewise rectified flow and demonstrates advantageous results in various personalization tasks.

## 2 Background

**Personalized image generation** studies incorporating user-specified subjects into the text-to-image generation pipeline. To preserve the subject's identity, the seminal works Textual Inversion (Gal et al., 2023) and DreamBooth (Ruiz et al., 2023a) finetune conditional embeddings or model parameters for each subject, which imposes high computational costs. Subsequent literature resorts to more efficient parameters (Hu et al., 2022; Han et al., 2023; Yuan et al., 2023) or a pre-trained subject encoder (Wei et al., 2023; Ye et al., 2023) to allow personalization within a few minutes or even without tuning. At the other end, a recent trend is the reuse of existing discriminators to improve identity consistency, such as extracting discriminative face features as the condition (Ye et al., 2023; Wang et al., 2024b) or as a training objective for the encoder (Peng et al., 2024; Gal et al., 2024; Guo et al., 2024). However, these models require extensive pre-training on domain-specific data, *e.g.* LAION-Face 50M (Zheng et al., 2022). In contrast, our method is a training-free approach that exploits existing discriminators based on the recent rectified flow model, allowing flexible personalization for a variety of tasks.

**Rectified flow** is an instance of flow-based generative models (Song et al., 2021; Xu et al., 2022; Liu et al., 2023a; Albergo and Vanden-Eijnden, 2023; Lipman et al., 2023). They aim to learn a velocity field $v$ that maps random noise $z_0 \sim \pi_0$ to samples from a complex distribution $z_1 \sim \pi_{\text{data}}$ via an ordinary differential equation (ODE):

$$d z_t = v(z_t, t) dt. \tag{1}$$

Instead of directly solving the ODE (Chen et al., 2018), rectified flow (Liu et al., 2023a) simply learns a linear interpolation between the two distributions by minimizing the following objective:

$$\min_{v} \int_0^1 \mathbb{E}\left[\|(z_1 - z_0) - v(z_t, t)\|^2\right] dt. \tag{2}$$

This procedure straightens the flow trajectory and thus allows faster sampling. Ideally, a well-trained rectified flow is a straight flow with uniform velocity $v(z_t, t) = v(z_0, 0)$ following:

$$z_t = z_0 + v(z_t, t) t. \tag{3}$$

Recently, rectified flow has shown promising efficiency (Liu et al., 2024b) and quality (Esser et al., 2024; Yan et al., 2024) in text-to-image generation. Our work extends its capabilities and theoretical properties to personalized image generation via classifier guidance.

**Classifier guidance,** initially proposed for class-conditioned diffusion models (Dhariwal and Nichol, 2021), introduces a test-time mechanism to adjust the predicted noise $\epsilon(z_t, t)$ based on the guidance from a classifier. Given condition $c$ and classifier output $p(c|z_t)$, the adjustment is formulated as:

$$\hat{\epsilon}(z_t, t) = \epsilon(z_t, t) + s \cdot \sigma_t \nabla_{z_t} \log p(c|z_t), \tag{4}$$

where $s$ denotes the guidance scale, and $\sigma_t$ is determined by the noise schedule. Noteworthy, the condition $c$ is not restricted to class labels, but can be extended to text (Nichol et al., 2022) and beyond. However, it is largely limited by the reliance on a noise-aware classifier for noised inputs $z_t$, which restricts the use of most pre-trained discriminators that only predict the likelihood $p(c|z_1)$ on clean images. Consequently, its usefulness is limited in practice. See Appendix B for more related work.

## 3 Method

This work aims at customizing rectified flow with classifier guidance. We show that the above limit of classifier guidance may be solved with a simple fixed-point solution for rectified flow (Section 3.1). To improve its stability, Section 3.2 proposes a new anchored classifier guidance with a convergence guarantee. Lastly, the implementation and applications are described in Sections 3.3 and 3.4.

### 3.1 Classifier Guidance for Rectified Flow

This section first derives the vanilla classifier guidance for rectified flow, and then present an initial attempt to remove the need for the noise-aware classifier $p(c|z_t)$, which is based on a new fixed-point solution of classifier guidance assuming that the rectified flow is ideally straight.

The classifier guidance can be derived as modifying the potential associated with the rectified flow. According to the Helmholtz decomposition, a velocity field $v$ may be decomposed into:

$$v(z_t, t) = \nabla_{z_t} \phi(z_t, t) + r(z_t, t), \tag{5}$$

where $\phi$ is a scaler potential and $r$ is a divergence-free rotation field. They can be determined by solving the Poisson's equation $\nabla^2 \phi(z_t, t) = \nabla \cdot v(z_t, t)$, but this is beyond our focus. We directly add a new potential proportional to the log-likelihood to simulate classifier guidance, as follows:

$$\hat{v}(z_t, t) = \nabla_{z_t}\left[\phi(z_t, t) + s \cdot \log p(c|z_t)\right] + r(z_t, t), \tag{6}$$

where $s$ denotes the guidance scale, and $\hat{}$ is used to distinguish the new flow from the original one. Subtracting the above two equations yields the vanilla classifier guidance, similar in form to Eq. (4):

$$\hat{v}(z_t, t) = v(z_t, t) + s \cdot \nabla_{z_t} \log p(c|z_t). \tag{7}$$

While this classifier guidance should allow for test-time conditioning of rectified flow, it cannot be applied in the absence of noise-aware classifier $p(c|z_t)$. In the following, we show that this limitation may be overcome by exploiting the straightness property of rectified flow.

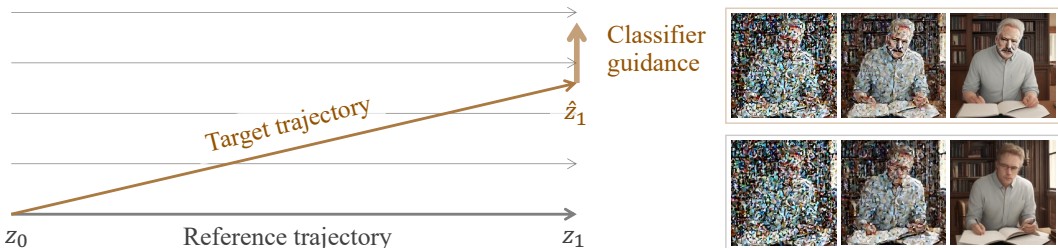

Figure 2: Illustration of anchored classifier guidance for rectified flow. Left: we propose to guide the flow trajectory while implicitly enforcing it to flow straight and stay close to a reference trajectory. Right: comparison of the new trajectory with the reference trajectory (in the last three sampling steps).

**Attempt to bypass noise-aware classifier.** We make a key observation that the intermediate classifier guidance $\nabla_{\boldsymbol{z}_t} \log p(c|\boldsymbol{z}_t)$ can be circumvented by approximating the new flow trajectory to be straight (an ideal guidance should preserve the properties of rectified flow) and focusing on the endpoint $\boldsymbol{z}_1$. Formally, substituting $t = 1$ in Eqs. (3) and (7) allows skipping any intermediate guidance terms:

$$\begin{aligned}
\boldsymbol{z}_1 &= \boldsymbol{z}_0 + \hat{\boldsymbol{v}}(\boldsymbol{z}_1, 1) \\
&= \boldsymbol{z}_0 + \boldsymbol{v}(\boldsymbol{z}_1, 1) + s \cdot \nabla_{\boldsymbol{z}_1} \log p(c|\boldsymbol{z}_1).
\end{aligned} \tag{8}$$

Interestingly, this turns out to be a fixed-point problem w.r.t. $\boldsymbol{z}_1$, suggesting that the classifier-guided flow trajectory could be solved iteratively by numerical methods such as the fixed-point iteration, without knowing the noise-aware classifier. This greatly enhances the flexibility of classifier guidance to a variety of off-the-shelf image discriminators. However, our further analysis reveals both empirical (Section 4.3) and theoretical evidence questioning the convergence of this iterative approach:

**Proposition 1.** *There exist Lipschitz continuous functions $\boldsymbol{v}(\boldsymbol{z}_1, 1)$ and $\nabla_{\boldsymbol{z}_1} \log p(c|\boldsymbol{z}_1)$, such that the fixed-point iteration for solving the target trajectory based on Eq. (8) is not guaranteed to converge by the Banach fixed-point theorem (Banach, 1922), irrespective of the choice of $s > 0$.*

*Proof.* Consider the following construction. Let $\boldsymbol{v}(\boldsymbol{z}_1, 1)$ and $\nabla_{\boldsymbol{z}_1} \log p(c|\boldsymbol{z}_1)$ be identical functions with a Lipschitz constant greater than 1. Then, the Lipschitz constant of the right-hand side of the fixed-point equation is greater than 1 for any $s > 0$. This violates the Banach fixed-point theorem's requirement for a Lipschitz constant strictly less than 1, thus convergence is not guaranteed. □

Proposition 1 shows that the derived fixed-point solution may not always be practical. Intuitively, even with a small perturbation at $\boldsymbol{z}_1$, the target flow trajectory estimated by Eq. (8) could diverge significantly after iterated updates, which hinders the controllability of rectified flow. This motivates us to anchor the target flow trajectory to a reference trajectory to stabilize its solving process.

## 3.2 Anchored Classifier Guidance

This section establishes a new type of classifier guidance based on a reference trajectory. The idea is to constrain the new trajectory to be straight and near the reference trajectory, as illustrated in Fig. 2. It provides a better convergence guarantee and a certain degree of interpretability.

Let $\hat{\boldsymbol{z}}_t$ and $\boldsymbol{z}_t$ represent two flow trajectories originating from the common starting point $\boldsymbol{z}_0$ with or without classifier guidance. The symbol ˆ denotes the new trajectory with classifier guidance. Assuming the two trajectories are close and straight (ideally preserving the characteristics of rectified flow), their difference can be estimated based on Eq. (7) and the first-order Taylor expansion:

$$\begin{aligned}
\hat{\boldsymbol{v}}(\hat{\boldsymbol{z}}_t, t) - \boldsymbol{v}(\boldsymbol{z}_t, t) &= \boldsymbol{v}(\hat{\boldsymbol{z}}_t, t) + s \cdot \nabla_{\hat{\boldsymbol{z}}_t} \log p(c|\hat{\boldsymbol{z}}_t) - \boldsymbol{v}(\boldsymbol{z}_t, t) \\
&\approx [\nabla_{\boldsymbol{z}_t} \boldsymbol{v}(\boldsymbol{z}_t, t)] (\hat{\boldsymbol{z}}_t - \boldsymbol{z}_t) + s \cdot \nabla_{\hat{\boldsymbol{z}}_t} \log p(c|\hat{\boldsymbol{z}}_t) \\
&= [\nabla_{\boldsymbol{z}_t} \boldsymbol{v}(\boldsymbol{z}_t, t) t] (\hat{\boldsymbol{v}}(\hat{\boldsymbol{z}}_t, t) - \boldsymbol{v}(\boldsymbol{z}_t, t)) + s \cdot \nabla_{\hat{\boldsymbol{z}}_t} \log p(c|\hat{\boldsymbol{z}}_t) \\
&= [\boldsymbol{I} - \nabla_{\boldsymbol{z}_t} \boldsymbol{z}_0] (\hat{\boldsymbol{v}}(\hat{\boldsymbol{z}}_t, t) - \boldsymbol{v}(\boldsymbol{z}_t, t)) + s \cdot \nabla_{\hat{\boldsymbol{z}}_t} \log p(c|\hat{\boldsymbol{z}}_t),
\end{aligned} \tag{9}$$

where the final step is derived from Eq. (3). From here, a new form of classifier guidance is obtained:

$$\hat{\boldsymbol{v}}(\hat{\boldsymbol{z}}_t, t) = \boldsymbol{v}(\boldsymbol{z}_t, t) + s \cdot [\nabla_{\boldsymbol{z}_0} \boldsymbol{z}_t] \nabla_{\hat{\boldsymbol{z}}_t} \log p(c|\hat{\boldsymbol{z}}_t). \tag{10}$$

This new classifier guidance anchors the target velocity to a predetermined reference velocity $\boldsymbol{v}(\boldsymbol{z}_t, t)$ that is dependent only on $t$ and irrelevant to the current state $\hat{\boldsymbol{z}}_t$, thereby constraining the target flow trajectory near the reference trajectory and improving its controllability. Next, we extend its applicability to the more common scenarios where the noise-aware classifier $p(c|\hat{\boldsymbol{z}}_t)$ is absent.

**Bypassing noise-aware classifier.** To circumvent the intermediate classifier guidance, we follow the previous practice of substituting $t = 1$ into Eqs. (3) and (10), yielding a fixed-point problem w.r.t. $\hat{\boldsymbol{z}}_1$:

$$\hat{\boldsymbol{z}}_1 = \boldsymbol{z}_1 + s \cdot [\nabla_{\boldsymbol{z}_0} \boldsymbol{z}_1] \nabla_{\hat{\boldsymbol{z}}_1} \log p(c|\hat{\boldsymbol{z}}_1). \tag{11}$$

As can be seen, the target endpoint $\hat{\boldsymbol{z}}_1$ is also anchored to a known reference point $\boldsymbol{z}_1$, which should enhance its stability in the solving process via fixed-point iteration or alternative numerical methods. Below, we exemplify its favorable theoretical property using the fixed-point iteration:

**Proposition 2.** *Suppose $\nabla_{\hat{\boldsymbol{z}}_1} \log p(c|\hat{\boldsymbol{z}}_1)$ is Lipschitz continuous w.r.t. $\hat{\boldsymbol{z}}_1$, the fixed-point iteration to solve the target trajectory by Eq. (11) exhibits at least linear convergence with a properly chosen $s$.*

*Proof.* Denote the Frobenius norm of $\nabla_{\boldsymbol{z}_0} \boldsymbol{z}_1$ as $L_1$, and the Lipschitz constant of $\nabla_{\hat{\boldsymbol{z}}_1} \log p(c|\hat{\boldsymbol{z}}_1)$ as $L_2$. The Lipschitz constant of the right side of the equation w.r.t. $\hat{\boldsymbol{z}}_1$ is upper bounded by $s \cdot L_1 \cdot L_2$. By choosing a sufficiently small $s < 1/(L_1 \cdot L_2)$, the Lipschitz constant of the right side is reduced to less than 1, thus ensuring linear convergence by the Banach fixed-point theorem. $\square$

**Interpretation of new classifier guidance.** In addition to the above convergence guarantee, our new classifier guidance can be interpreted by connecting with gradient backpropagation. From Eq. (10) one could obtain an estimate of the intermediate classifier guidance (see Appendix A for derivation):

$$\nabla_{\hat{\boldsymbol{z}}_t} \log p(c|\hat{\boldsymbol{z}}_t) = [\nabla_{\boldsymbol{z}_t} \boldsymbol{z}_1] \nabla_{\hat{\boldsymbol{z}}_1} \log p(c|\hat{\boldsymbol{z}}_1). \tag{12}$$

This suggests that our method is secretly estimating the intermediate classifier guidance with gradient backpropagation. While this is implicitly assumed or directly used in recent works that adapt classifier guidance to flow-based models (Wallace et al., 2023; Liu et al., 2023b; Ben-Hamu et al., 2024), it is explicitly derived here based on a very different assumption (the straightness of the flow trajectory). Such a connection helps to rationalize both our adopted assumption and the existing practice.

## 3.3 Practical Algorithm

**Extension to piecewise rectified flow.** The above analyses are performed based on the assumption that the rectified flow is well-trained and straight, which is often not the case in reality. In fact, existing rectified flow usually require multiple sampling steps due to the inherent curvature in the flow trajectory. Inspired by Yan et al. (2024), we adopt a relaxed assumption during implementation that the rectified flow is piecewise linear. Let there be $K$ time windows $\{[t_k, t_{k-1})\}_{k=K}^1$ where $1 = t_K > \cdots > t_k > t_{k-1} > \cdots > t_0 = 0$, and the flow trajectory is assumed straight within each time window, then the inference procedure can be expressed as:

$$\boldsymbol{z}_t = \boldsymbol{z}_{t_{k-1}} + \boldsymbol{v}(\boldsymbol{z}_t, t)(t - t_{k-1}), \tag{13}$$

where $k$ is the index of the time window $[t_k, t_{k-1})$ that $t$ belongs to. Note that this framework is also compatible with the vanilla rectified flow by setting $K$ to the number of sampling steps.

The previously derived fixed-point iteration in Eq. (11) cannot be applied directly, since its assumption that the target and reference trajectory segments share the same starting point (e.g. $\hat{\boldsymbol{z}}_{t_{k-1}} = \boldsymbol{z}_{t_{k-1}}$) may be violated after updates. A quick fix is to reinitialize the reference trajectory every round with predictions for updated target starting points. This allows to formulate the following problem:

$$\begin{aligned} \hat{\boldsymbol{z}}_{t_k} &= \boldsymbol{z}_{t_k}^e + s \cdot \left[ \nabla_{\boldsymbol{z}_{t_{k-1}}} \boldsymbol{z}_{t_k}^e \right] \nabla_{\hat{\boldsymbol{z}}_{t_k}} \log p(c|\hat{\boldsymbol{z}}_{t_k}) \\ &= \boldsymbol{z}_{t_k}^e + s \cdot \left[ \nabla_{\boldsymbol{z}_{t_{k-1}}} \boldsymbol{z}_1^e \right] \nabla_{\hat{\boldsymbol{z}}_1} \log p(c|\hat{\boldsymbol{z}}_1), \end{aligned} \tag{14}$$

where the last step is obtained by recursively applying Eq. (12) to backpropagate the guidance signal, and a superscript $e$ is introduced to denote the endpoint of the previous trajectory segment, as the above fix may disconnect different segments of the reference trajectory. Meanwhile, a straight-through estimator (Bengio et al., 2013) is applied to allow computing the Jacobian across different trajectory segments by estimating the Jacobian between the adjacent points $\boldsymbol{z}_{t_k}$ and $\boldsymbol{z}_{t_k}^e$ with $\boldsymbol{I}$.

---
**Algorithm 1** Anchored Classifier Guidance
---
**Input:** rectified flow $v$, classifier $p(c|\cdot)$, sampling steps $K$, iterations $N$.

    Initialize reference trajectory $z_{t_k}^{[0]}$ from $v$.                                     ▷ Eq. (13)

    Initialize target trajectory $\hat{z}_{t_k}^{[0]} \leftarrow z_{t_k}^{[0]}$.

    **for** $i \leftarrow 0$ to $N-1$ **do**

        Update reference trajectory with predicted starting points $z_{t_k}^{[i+1]}$.           ▷ Eq. (15)

        Update target trajectory $\hat{z}_{t_k}^{[i+1]}$ with classifier output $p(c|\hat{z}_1^{[i]})$.       ▷ Eq. (16)

**Output:** target trajectory $\hat{z}_{t_k}^{[N]}$ subject to condition $c$.
---

**Solving target flow trajectory.** The target trajectory under classifier guidance, subject to Eq. (14), can be estimated iteratively by starting with $\hat{z}_{t_k}^{[0]} = z_{t_k}^{[0]}$ and performing the following iterations:

$$z_{t_k}^{[i+1]} = z_{t_k}^{[i]} + \underbrace{z_{t_k}^{e[i+1]} - z_{t_k}^{e[i]}}_{\text{current offset}} + \underbrace{\hat{z}_{t_k}^{[i]} - z_{t_k}^{e[i]}}_{\text{predicted update}}, \tag{15}$$

$$\hat{z}_{t_k}^{[i+1]} = z_{t_k}^{e[i+1]} + s \cdot \left[ \nabla_{z_{t_{k-1}}^{[i+1]}} z_1^{e[i+1]} \right] \nabla_{\hat{z}_1^{[i]}} \log p(c|\hat{z}_1^{[i]}), \tag{16}$$

where the superscript $[i]$ is used to indicate the target and reference trajectories at the $i$-th iteration. Specifically, Eq. (15) implements the prediction of updated target starting points by extrapolating from history updates, and Eq. (16) tackles the derived problem. Note that there are more sophisticated methods for predicting target starting points and solving this problem, *e.g.* quasi-Newton methods, but we opt for simplicity here and leave their exploration to future work. The complete procedure for implementing the proposed classifier guidance is summarized by Algorithm 1.

### 3.4 Applications

The proposed algorithm is flexible for various personalized image generation tasks on human faces and common subjects. Given a reference image $z_{\text{ref}}$ and our generated image $\hat{z}_1$, we use their feature similarity on an off-the-shelf discriminator $f$, *e.g.* the face specialist ArcFace (Deng et al., 2019) or a self-supervised backbone DINOv2 (Oquab et al., 2023), as classifier guidance. In addition, to improve the guidance signal, a face detector or an open-vocabulary object detector $g$ is employed to locate the identity-relevant region for feature extraction. Formally, the classifier output is as follows:

$$p(c|\hat{z}_1^{[i]}) = \text{sim}\left( f \circ g(\hat{z}_1^{[i]}), f \circ g(z_{\text{ref}}) \right). \tag{17}$$

More details are described in Appendix C. Notably, both configurations can be flexibly extended to a multi-subject scenario by incorporating a bipartite matching step between multiple detected subjects.

## 4 Experiments

### 4.1 Experimental Settings

**Datasets.** Our method does not involve training data, as it operates only at test time. For face-centric evaluation, we follow Pang et al. (2024) to evaluate on 20 prompts with the first 200 images from CelebA-HQ (Liu et al., 2015; Karras et al., 2018) as reference images. For subject-driven generation, we conduct qualitative studies on a subset of examples from the DreamBooth dataset (Ruiz et al., 2023b), spanning 10 subjects across two live subject categories and three object categories.

**Metrics.** Three metrics are considered: identity similarity, prompt consistency, and computation time. The first two are measured using an ArcFace model (Deng et al., 2019) and CLIP encoders (Radford et al., 2021), while the latter is tested on an NVIDIA A800 GPU. We reproduce the latest methods IP-Adapter (Ye et al., 2023), PhotoMaker (Li et al., 2024), and InstantID (Wang et al., 2024b) for a comprehensive comparison, and also include the existing baselines in Pang et al. (2024).

**Implementation details.** We experiment with a frozen piecewise rectified flow (Yan et al., 2024) finetuned from Stable Diffusion 1.5 (Rombach et al., 2022) with 4 equally divided time windows. The number of sampling steps is set to a minimum $K = 4$ given the memory overhead of backpropagation.

Table 1: Quantitative comparison for face-centric personalization. The inference time is measured in seconds on an NVIDIA A800. Unlike the previous state-of-the-art methods that require training on large face datasets (the number of images is listed for reference), our method achieves superior performance in a training-free manner, by exploiting the guidance from an off-the-shelf discriminator.

| Method | Base model | Training | Identity ↑ | Prompt ↑ | Time ↓ |
|---|---|---|---|---|---|
| Textual Inversion (Gal et al., 2023) | SD 2.1 | - | 0.2115 | 0.2498 | 6331 |
| DreamBooth (Ruiz et al., 2023a) | SD 2.1 | - | 0.2053 | 0.3015 | 623 |
| NeTI (Alaluf et al., 2023) | SD 1.4 | - | 0.3789 | 0.2325 | 1527 |
| Celeb Basis (Yuan et al., 2023) | SD 1.4 | - | 0.2070 | 0.2683 | 140 |
| Cross Initialtion (Pang et al., 2024) | SD 2.1 | - | 0.2517 | 0.2859 | 346 |
| IP-Adapter (Ye et al., 2023) | SD 1.5 | 10M | 0.4778 | 0.2627 | **2** |
| PhotoMaker (Li et al., 2024) | SDXL | 112K | 0.2271 | 0.3079 | 4 |
| InstantID (Wang et al., 2024b) | SDXL | 60M | 0.5806 | 0.3071 | 6 |
| RectifID (20 iterations) | SD 1.5 | - | 0.4860 | 0.2995 | 9 |
| RectifID (100 iterations) | SD 1.5 | - | **0.5930** | 0.2933 | 46 |
| RectifID (20 iterations) | SD 2.1 | - | 0.5034 | **0.3151** | 20 |

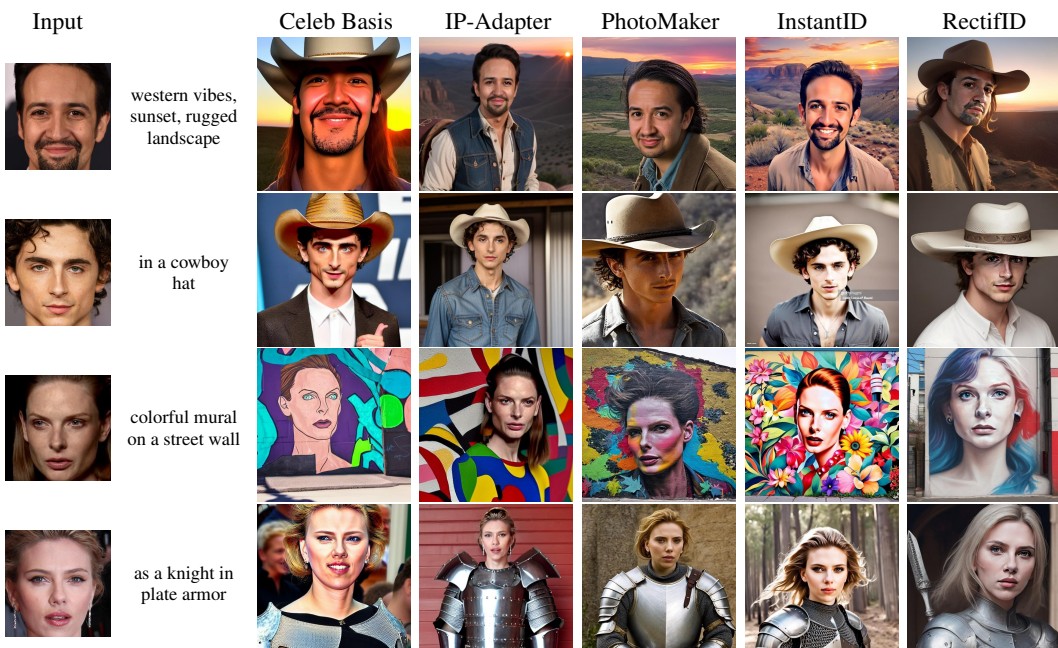

Figure 3: Qualitative comparison for face-centric personalization. See Figs. 9 to 12 for more samples.

A naive implementation takes 14GB of GPU memory, which fits on a range of consumer-grade GPUs. More results on alternative rectified flows can be found in Appendix D.4. For hyperparameters, the guidance scale is fixed to $s = 1$ in quantitative evaluation. Meanwhile, for stability, the gradient is normalized following Karunratanakul et al. (2024). The number of iterations is set to $N = 100$.

## 4.2 Main Results

**Face-centric personalization.** Table 1 and Fig. 3 compare our method (denoted RectifID) with extensive baselines. Overall, our training-free approach achieves state-of-the-art performance in quantitative evaluations. Specifically, we observe that: (1) our SD 1.5-based implementation yields the highest identity similarity of all, and leads in prompt consistency among SD 1.x-based methods. It is also computationally efficient, *e.g.* taking less time than existing tuning-based methods, and outperforming the training-based IP-Adapter (Ye et al., 2023) in a near inference time of 9 seconds vs. 2 seconds. (2) By simply replacing the base diffusion model with SD 2.1 at its default image size, our prompt consistency further surpasses SDXL (Podell et al., 2024)-based models. Note, however,

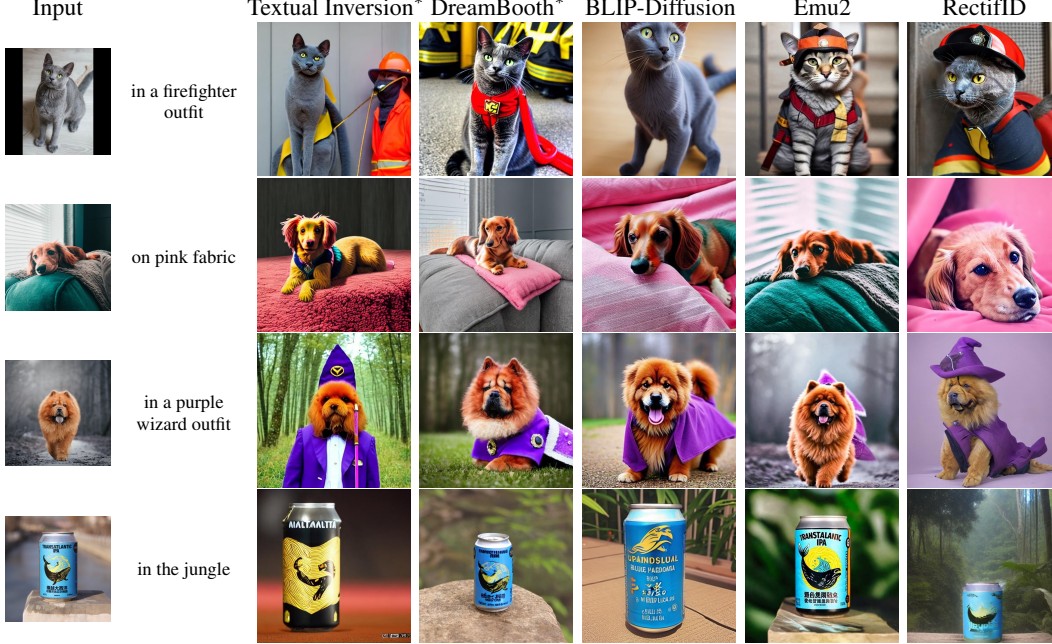

Figure 4: Qualitative comparison for subject-driven generation. * denotes finetuned with multiple images of the target subject to achieve sufficient identity consistency. See Fig. 13 for more samples.

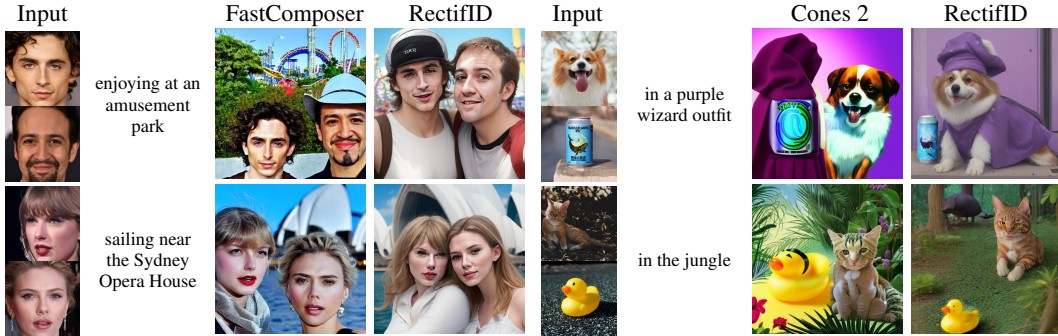

Figure 5: Qualitative comparison for multi-subject personalization. See Fig. 14 for more samples.

that the rest of this paper still uses SD 1.5 for a fair comparison to SD 1.x-based baselines in various personalization tasks, excluding potential improvements from using better base models. (3) In general, our method takes a big step towards bridging the substantial performance gap with training-based personalization methods by exploring the effectiveness of training-free classifier guidance.

For face-centric qualitative comparison in Fig. 3, our method remains advantageous as its generated images by the guidance of the face discriminator exhibit high identity consistency. In comparison, InstantID (Wang et al., 2024b) delivers a near level of consistency by controlling face landmarks, but sometimes distorts the face shape (the first and third images) and contains much less natural variation. More generated samples are provided in Figs. 11 and 12 in the appendix.

**Subject-driven generation.** Our approach is flexibly extended beyond human faces towards more subjects, including certain common animals and regularly shaped objects. To validate our flexibility, Fig. 4 qualitatively compares it on three cats or dogs and a regularly shaped can, where the images generated by our method achieve highly competitive identity and prompt consistency. In comparison, the state-of-the-art method Emu2 (Sun et al., 2024), as a generalist multimodal large language model, yields high identity similarity largely by reconstructing the input image, which limits its usefulness. The tuning-based Textual Inversion (Gal et al., 2023) and DreamBooth (Ruiz et al., 2023a) only work well with multiple images and exhibit inferior prompt consistency due to finetuned model parameters or prompt embeddings. See Fig. 13 in the appendix for additional results from more subjects.

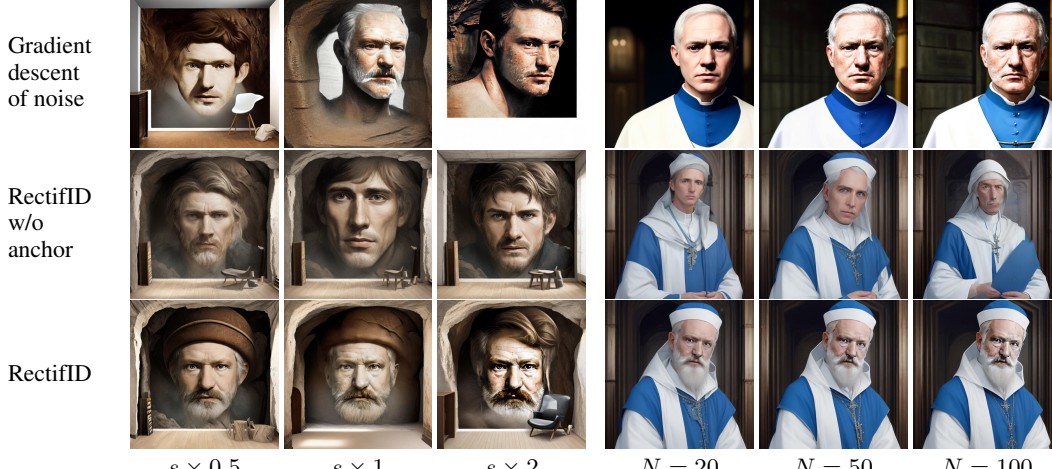

| | | | | | |
|---|---|---|---|---|---|
| $s \times 0.5$ | $s \times 1$ | $s \times 2$ | $N = 20$ | $N = 50$ | $N = 100$ |

Figure 6: Comparison with alternative designs at varying guidance scale (or learning rate) and iterations. The prompts are "cave mural depicting a person" and "a person as a priest in blue robes". The base learning rate for gradient descent is 0.4, with momentum of 0.9 and an $\ell_2$ regularizer of 1.0.

**Multi-subject personalization.** Our method can be further extended to multi-subject scenarios via a bipartite matching step. Figure 5 compares it to the domain experts FastComposer (Xiao et al., 2023) and Cones 2 (Liu et al., 2023c) on composing multiple faces, live subjects and objects. As can be seen, our method achieves overall advantageous identity consistency, in spite of differences in non-persistent attributes such as hairstyle. Image semantics and quality are also well preserved, as exemplified by the amusement park details in the first image, with some others even surpassing the SD 2.1-based specialized model Cones 2. More generated samples can be found in Fig. 14 in the appendix.

## 4.3 Ablation Study

To justify the effectiveness of our proposed classifier guidance, Fig. 6 and Table 2 compare it with two variants: the previously derived guidance without anchor, namely using Eq. (8), and a gradient descent method on the initial noise similar to DOODL (Wallace et al., 2023) and D-Flow (Ben-Hamu et al., 2024). The figure depicts that the gradient descent is unstable (left) and converges relatively slowly (right) despite using momentum and $\ell_2$ regularization.

Table 2: Quantitative comparison with alternative designs. The number of iterations is 100, and the remaining settings for gradient descent follow Fig. 6.

| Method | Identity ↑ | Prompt ↑ |
|---|---|---|
| Gradient descent of noise | 0.5249 | 0.2842 |
| RectifID w/o anchor | 0.1158 | 0.2916 |
| RectifID | **0.5930** | **0.2933** |

And its identity preservation is sensitive to the learning rate. Though our new fixed-point formulation allows for a more stable layout, the initial version fails to converge as the face feature keeps drifting. In contrast, our full method exhibits better stability (left) and faster convergence (right) by implicitly regularizing the flow trajectory to be close and straight. This is further supported by the quantitative comparison, where our method delivers better identity and prompt consistency than the alternatives. Further analysis for hyperparameter sensitivity is provided in Appendix D.3.

## 4.4 Generalization

To validate the generalizability of our approach to broader application scenarios, we have extended it to more controllable generation tasks by directly using the guidance functions from Universal Guidance (Bansal et al., 2024). The experimental results under the guidance of segmentation map or style image are illustrated in Fig. 7. As shown, our classifier guidance can perform both tasks without any additional tuning, faithfully following the various forms of control signals provided by the user. This confirms the adaptability of our approach for various controllable generation tasks. Additional generalization analysis of our method for broader diffusion models is presented in Appendix D.1.

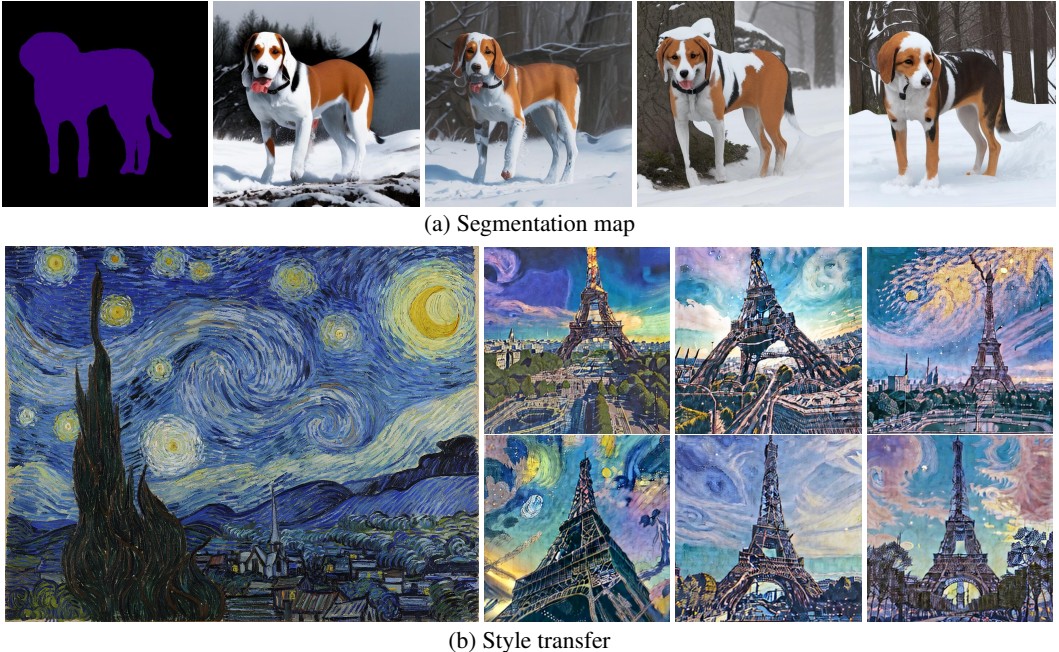

(a) Segmentation map

(b) Style transfer

Figure 7: Experimental results for more controllable generation tasks. The first column shows the guidance, and the rest are the generated results. Our method is extended to various controllable generation tasks by incorporating the guidance functions from Universal Guidance (Bansal et al., 2024).

# 5    Conclusion

This work presents a training-free personalized image generation method using anchored classifier guidance. It extends the applicability of the original classifier guidance based on two key findings: first, by developing on a rectified flow framework assuming ideal straightness, the classifier guidance can be transformed into a new fixed-point formulation involving only clean image-based discriminators; secondly, anchoring the flow trajectory to a reference trajectory greatly improves its solving stability. The derived anchored classifier guidance allows flexible reuse of existing image discriminators to improve identity consistency, as validated by extensive experiments on various personalized image generation tasks for human faces, live subjects, certain objects, and multiple subjects.

**Acknowledgement:** This research work is supported by National Key R&D Program of China (No. 2022ZD0160305), a research grant from China Tower Corporation Limited, an internal grant (No. 2024JK28) and a grant from Beijing Aerospace Automatic Control Institute.

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

# A Detailed Derivations

Most of the equations in our paper are accompanied by their derivations, except for Eqs. (11) and (12). For the sake of completeness, their derivations are supplemented here.

Equation (11): We substitute $t = 1$ into Eqs. (3) and (10) to obtain:

$$\begin{aligned}
\hat{z}_1 &= z_0 + \hat{v}(\hat{z}_1, 1) \\
&= z_0 + v(z_1, 1) + s \cdot [\nabla_{z_0} z_1] \nabla_{\hat{z}_1} \log p(c|\hat{z}_1) \\
&= z_1 + s \cdot [\nabla_{z_0} z_1] \nabla_{\hat{z}_1} \log p(c|\hat{z}_1).
\end{aligned} \tag{18}$$

Equation (12): By applying Eq. (10) twice and utilizing the straightness property of rectified flow, we have:

$$\begin{aligned}
\nabla_{\hat{z}_t} \log p(c|\hat{z}_t) &= 1/s \cdot [\nabla_{z_t} z_0] \left( \hat{v}(\hat{z}_t, t) - v(z_t, t) \right) \\
&= 1/s \cdot [\nabla_{z_t} z_0] \left( \hat{v}(\hat{z}_1, 1) - v(z_1, 1) \right) \\
&= [\nabla_{z_t} z_0] [\nabla_{z_0} z_1] \nabla_{\hat{z}_1} \log p(c|\hat{z}_1) \\
&= [\nabla_{z_t} z_1] \nabla_{\hat{z}_1} \log p(c|\hat{z}_1).
\end{aligned} \tag{19}$$

# B Related Work on Classifier Guidance

Since the proposal of classifier guidance (Dhariwal and Nichol, 2021) which uses a special noise-aware classifier as training-free guidance for diffusion models, many efforts have been made in order to extend its applicability to off-the-shelf loss guidance. They can be grouped into three categories: (1) Early literature focuses on simpler objectives for linear inverse problems such as image super-resolution, deblurring, and inpainting (Chung et al., 2022, 2023; Wang et al., 2023; Zhu et al., 2023). (2) These methods can be extended to more complex discriminators through various approximations. Yu et al. (2023); Song et al. (2023) use Tweedie's formula and Monte Carlo method, respectively, to estimate the integrated classifier guidance. Bansal et al. (2024); He et al. (2024) perform updates directly in the clean data space, with the latter imposing an additional manifold constraint. Similarly, Gaussian spherical constraint is explored in Yang et al. (2024). Mou et al. (2024) advance these techniques to more versatile editing tasks. The above methods are further unified in Ye et al. (2024). (3) A recent line of work directly uses gradient descent with specific diffusion models. To enable their gradient computation, DiffusionCLIP (Kim et al., 2022a) relies on shortened ODE trajectories, while FlowGrad (Liu et al., 2023b) adopts a non-uniform ODE discretization and decomposed gradient computation. DOODL (Wallace et al., 2023), DNO (Karunratanakul et al., 2024) and D-Flow (Ben-Hamu et al., 2024) use invertible models or flow models to backpropagate gradient to the initial noise. Our proposed method is related to the third category, focusing on rectified flow whose approximation remains understudied. Moreover, it features a fixed-point formulation with a convergence guarantee for ideal rectified flow, which allows potentially better stability over the existing approaches, *e.g.* gradient descent on initial noise, as empirically validated in Section 4.3.

Recent studies also explore the use of pre-trained classifiers to guide personalized image generation, but mostly during model training. DiffFace (Kim et al., 2022b) and PhotoVerse (Chen et al., 2023) directly apply a face discriminator to noised images to compute an identity loss. PortraitBooth (Peng et al., 2024) improves loss quality by computing it at less noisy stages. More relevant to our work, LCM-Lookahead (Gal et al., 2024) and PuLID (Guo et al., 2024) utilize distilled diffusion models to generate images in few steps, allowing for direct gradient backpropagation of image-space losses. However, these personalization methods must first be trained on extensive face recognition data, *e.g.* LAION-Face 50M (Zheng et al., 2022). In comparison, we harness off-the-shelf face discriminators at test time based on the methodology of classifier guidance, enabling more flexible customization without training. And it can be generalized to other subjects by simply replacing the classifier.

# C Experimental Settings

**Method.** We mainly experiment on the recently proposed piecewise rectified flow (Yan et al., 2024). The model is finetuned from Stable Diffusion 1.5 (Rombach et al., 2022) on the LAION-Aesthetic-5+ dataset (Schuhmann et al., 2022) without special pre-training on human faces or subjects. We adopt a fixed image size of $512 \times 512$, number of sampling step $K = 4$, and a classifier-free guidance scale

of 3.0 during quantitative evaluation. The newly incorporated anchored classifier guidance uses a default guidance scale $s = 1.0$ and number of iterations $N = 100$. For qualitative studies, a few results are generated with a slightly different guidance scale $s = 0.5$ or $2.0$ for better visual quality or identity consistency. But in general, our method is not very sensitive to these hyperparameters. In terms of computational and memory overhead, our method takes less than 0.5s per iteration on an NVIDIA A800 GPU and fits on consumer-grade GPUs such as NVIDIA RTX 4080, the latter of which may be further improved with gradient checkpointing or the techniques in Liu et al. (2023b).

For face-centric personalization, our method is implemented with the antelopev2 model pack from the InsightFace library.[2] Specifically, it detects and crops the face regions with an SCRFD model (Guo et al., 2022), and then extracts face features using an ArcFace model (Deng et al., 2019) trained on Glint360K (An et al., 2021), consistent with most personalization methods that use a face model (Ye et al., 2023; Wang et al., 2024b; Gal et al., 2024). The resulting face feature is compared with the reference image to compute a cosine loss, which serves as the classifier guidance signal.

For subject-driven generation, we use an open-vocabulary object detector OWL-ViT (Minderer et al., 2022) to detect the region of interest, and extract visual features with DINOv2 (Oquab et al., 2023). The extracted feature is compared with the reference to calculate a cosine loss as the guidance signal. While this guidance already works well for various live subjects including multiple dogs and cats, we add an optional $\ell_1$ loss with a coefficient of 10.0 to help preserve the identity of certain objects, such as cans, vases and duck toys. The current implementation is still limited in not capturing the details of some irregularly shaped objects, *e.g.* plush toys, and we expect to resolve this issue with an improved discriminator, or by combining with existing image prompt techniques (Ye et al., 2023).

For multi-subject personalization, we consider a simplified case of exactly two subjects and perform the following: first detect the two subjects, then enumerate all possible bipartite matches with the reference subjects to minimize the matching cost. For more complex scenarios, a possible workaround is to formulate it as an quadratic assignment problem and apply efficient solvers (Tan and Mu, 2024).

**Evaluation.** For face-centric personalization, we follow Pang et al. (2024) in evaluating on the first 200 images in the CelebA-HQ dataset (Liu et al., 2015; Karras et al., 2018) with 20 text prompts including 15 realistic prompts and 5 stylistic prompts. The evaluation process reuses the code from Celeb Basis (Yuan et al., 2023),[3] which first detects the face region using a PIPNet (Jin et al., 2021) with a threshold of 0.5 and then computes the cosine similarity on face features extracted by an ArcFace model (Deng et al., 2019). It should be noted that our method adopts a different face detector and different alignment and cropping methods, so it does not overfit the evaluation protocol. For the baselines, in addition to those compared in Pang et al. (2024), we also evaluate the recently proposed IP-Adapter (Ye et al., 2023), PhotoMaker (Li et al., 2024), and InstantID (Wang et al., 2024b) on their recommended settings. The number of their sampling steps is set to 30 for a fair comparison. The checkpoint version of IP-Adapter is ip-adapter-full-face_sd15. The image size is set to $512 \times 512$ for IP-Adapter and $1024 \times 1024$ for PhotoMaker and InstantID based on SDXL (Podell et al., 2024). In the qualitative analysis, we also include Celeb Basis (Yuan et al., 2023) as a baseline method.

For subject-driven generation, we perform a qualitative rather than a quantitative comparison on a subset of the DreamBooth dataset (Ruiz et al., 2023a) due to the previously mentioned limitations. Nevertheless, many subjects are considered during the qualitative study, including 7 live subjects from two categories (cats and dogs) and 3 regularly shaped objects from different categories (cans, vases, and teapots). For the baselines, we incorporate Textual Inversion (Gal et al., 2023), DreamBooth (Ruiz et al., 2023a), BLIP-Diffusion (Li et al., 2023), and Emu2 (Sun et al., 2024) for extensive comparison. Their diffusion models and hyperparameter settings follow the official or Diffusers implementation.[4]

**Licenses.** The piecewise rectified flow (Yan et al., 2024) used in the main experiments is released under the BSD-3-Clause License and the 2-rectified flow (Liu et al., 2023a, 2024b) used in Appendix D.4 is released under the MIT License. The InsightFace library for face detection and recognition is released under the MIT License, while its pre-trained models are available for non-commercial research purposes only. The OWL-ViT (Minderer et al., 2022) and DINOv2 (Oquab et al., 2023) models for object detection and feature extraction are released under the Apache-2.0 License. For evaluation, the code of Celeb Basis (Yuan et al., 2023) is licensed under the MIT License. The

---

[2] https://github.com/deepinsight/insightface

[3] This is mentioned at https://github.com/lyuPang/CrossInitialization#metrics.

[4] For example, we use https://huggingface.co/docs/diffusers/training/text_inversion.

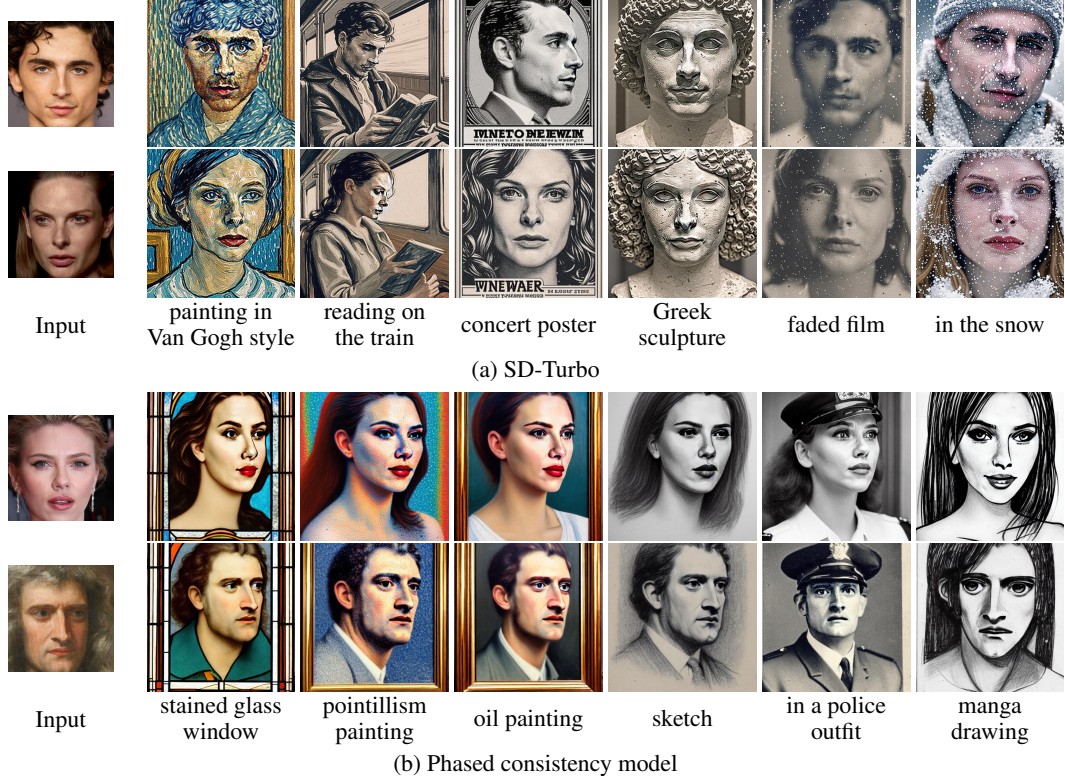

| Input | painting in Van Gogh style | reading on the train | concert poster | Greek sculpture | faded film | in the snow |

(a) SD-Turbo

| Input | stained glass window | pointillism painting | oil painting | sketch | in a police outfit | manga drawing |

(b) Phased consistency model

Figure 8: Generalization to few-step diffusion models, including SD-Turbo (Sauer et al., 2024) and phased consistency model (Wang et al., 2024a), both distilled from SD and using 4 sampling steps in inference. The results show that our method is effective for personalizing broader diffusion models.

CelebA-HQ (Liu et al., 2015; Karras et al., 2018) and DreamBooth (Ruiz et al., 2023a) datasets for quantitative and qualitative evaluation are released under the CC BY-NC 4.0 and CC-BY-4.0 licenses.

# D  Additional Results

## D.1  Generalization

While our classifier guidance is derived based on rectified flow, the same idea can be generalized to some few-step diffusion models by assuming straightness of their trajectories within each time step. We empirically demonstrate this in Fig. 8 with two popular few-step diffusion models, SD-Turbo (Sauer et al., 2024) and phased consistency model (Wang et al., 2024a). As the results indicate, our method effectively personalizes these diffusion models to generate identity-preserving images. We will continue to explore this approach for other generative models in future research.

## D.2  More Visualizations

More examples of our generated image are provided in Figs. 9 to 14. For face-centric personalized image generation, it is shown that our method can follow a variety of text prompts to generate both realistic or stylistic images while preserving the user-specified identity from a diverse group of people. Although there exist minor differences in the person's age and hairstyles, the face looks very similar to the reference image. In particular, the method demonstrates good generalizability among different piecewise rectified flows based on SDXL (Podell et al., 2024) and SD 1.5 (Rombach et al., 2022). For subject-driven generation, new results are presented from additional subject types, including different breeds of cats and dogs, and some regularly shaped objects such as vases, demonstrating the flexibility of our approach across different use cases. For multi-subject generation, it naturally blends multiple human faces or live subjects into a single image while maintaining the visual quality

and semantics, revealing a wider range of potentially interesting applications. Overall, our method demonstrates to be effective and flexible for various personalization tasks.

### D.3 Ablation Study

In addition to the ablation experiments in the main paper, we perform a sensitivity analysis of the two hyperparameters in our method, namely the guidance scale $s$ and the number of iterations $N$. Specifically, we study the effect of different $s$ under a fixed $N = 20$ and then the effect of different $N$ under a fixed $s = 1.0$. The results are summarized in Fig. 15. As can be observed, increasing both hyperparameters from 0 leads to a significant improvement in identity consistency, confirming the effectiveness of our proposed classifier guidance. Also, the performance is stable over a fairly wide range of hyperparameters, indicating that our approach is not very sensitive to hyperparameters. Furthermore, we find that the use of a small classifier guidance scale is actually beneficial for prompt consistency, possibly because it enhances the visual features, as demonstrated in Fig. 2.

### D.4 Experiments with 2-Rectified Flow

Figures 16 and 17 present additional qualitative results on a vanilla 2-rectified flow (Liu et al., 2023a, 2024b) finetuned on Stable Diffusion 1.4 (Rombach et al., 2022). As can be seen, our method continues to deliver satisfactory identity preservation when moving to a different model, despite a noticeable drop in the generation quality. Concretely, it integrates target subjects with some quite challenging prompts, such as a person swimming or getting a haircut, while showing very little interference with the original background, *e.g.* jungles and cityscapes. These results clearly validate the effectiveness of our proposed method in alternative rectified flow models.

Note that we also experimented with $K = 1$ on a single-step InstaFlow (Liu et al., 2024b) distilled from this 2-rectified flow, but found that it tended to converge to slightly distorted images. This may be attributed to the larger modeling error inherent in InstaFlow's distillation process, which reduces the effectiveness of our approach assuming each flow trajectory segment is well-trained and straight.

## E  Broader Impacts and Limitations

**Broader impacts.** The proposed method can be integrated with the emerging rectified flow-based models to enhance identity preservation and versatile control over existing AI art production pipelines. However, as a training-free personalization technique, it may increase the risk of image faking and have negative societal effects. Some immediate remedies include text-based prompt filters and AI-generated image detection, but it remains an open problem for a more principled solution, for which we advocate further research on data watermarking and model unlearning as potential mitigations. To further clarify it, we provide a more detailed explanation of these mitigations below:

- Prompt filtering, model unlearning: Since our method keeps the diffusion model intact, existing techniques for regulating diffusion models can be applied seamlessly, including prompt filters or unlearning methods. The former can be applied explicitly like the text filters in SD models, or implicitly via CFG as in Schramowski et al. (2023). The latter approaches involve finetuning the diffusion model to remove the ability to generate harmful content (Gandikota et al., 2023; Kumari et al., 2023).
- Data watermarking: To prevent misuse of personal images, one could add a protective watermark to their images (Van Le et al., 2023; Liu et al., 2024a). With this watermark or perturbation, the image can no longer be learned by common personalization methods. However, it is unclear how robust these watermarks are to training-free methods such as ours. An alternative watermarking scheme is to embed special watermark to the images generated by our proposed model, which would be invisible to the users yet identifiable by us (i.e., the tech provider). Images with such watermarks will be marked as being artificial.
- AI-generated image detection: As a post-hoc safety measure, it helps to distinguish fake images generated by the attackers from real images. Beyond above watermark-based scheme, more sophisticated data-driven methods have attracted increasing interest from the AI community. Despite that current methods still lack accuracy, we believe that developing reliable and widely available AI-generated image detectors is an important research direction.

**Limitations.** Our theoretical guarantee is limited to ideal rectified flow and cannot be generalized to more complex flow-based models. Empirically, anchoring the new flow trajectory to a reference trajectory only proves effective for faces, live subjects and certain regularly shaped objects, and remains insufficient for many objects with large structural variations, *e.g.* plush toys. Furthermore, while our method is training-free, its inference time has yet to match several training-based baselines, which may be addressed by applying more advanced numerical solvers to the derived problem.

Another important issue is the lack of pre-trained discriminators. To address this in the short term, we suggest first training a specialized discriminator and then applying our classifier guidance. There are two reasons for doing this instead of finetuning the generator directly: (1) training/finetuning a discriminator is usually more efficient and stable than training/finetuning a generator; (2) it can take full advantage of domain images that have no captions or even labels by using standard contrastive learning loss. In the future, scaling up vision-language models may be a general solution for these domains. The current models such as GPT-4V (OpenAI, 2023) have demonstrated certain generalizability across visual understanding tasks. As they continue to improve in generalizability and robustness, they will become a viable source for guiding diffusion models in new domains.

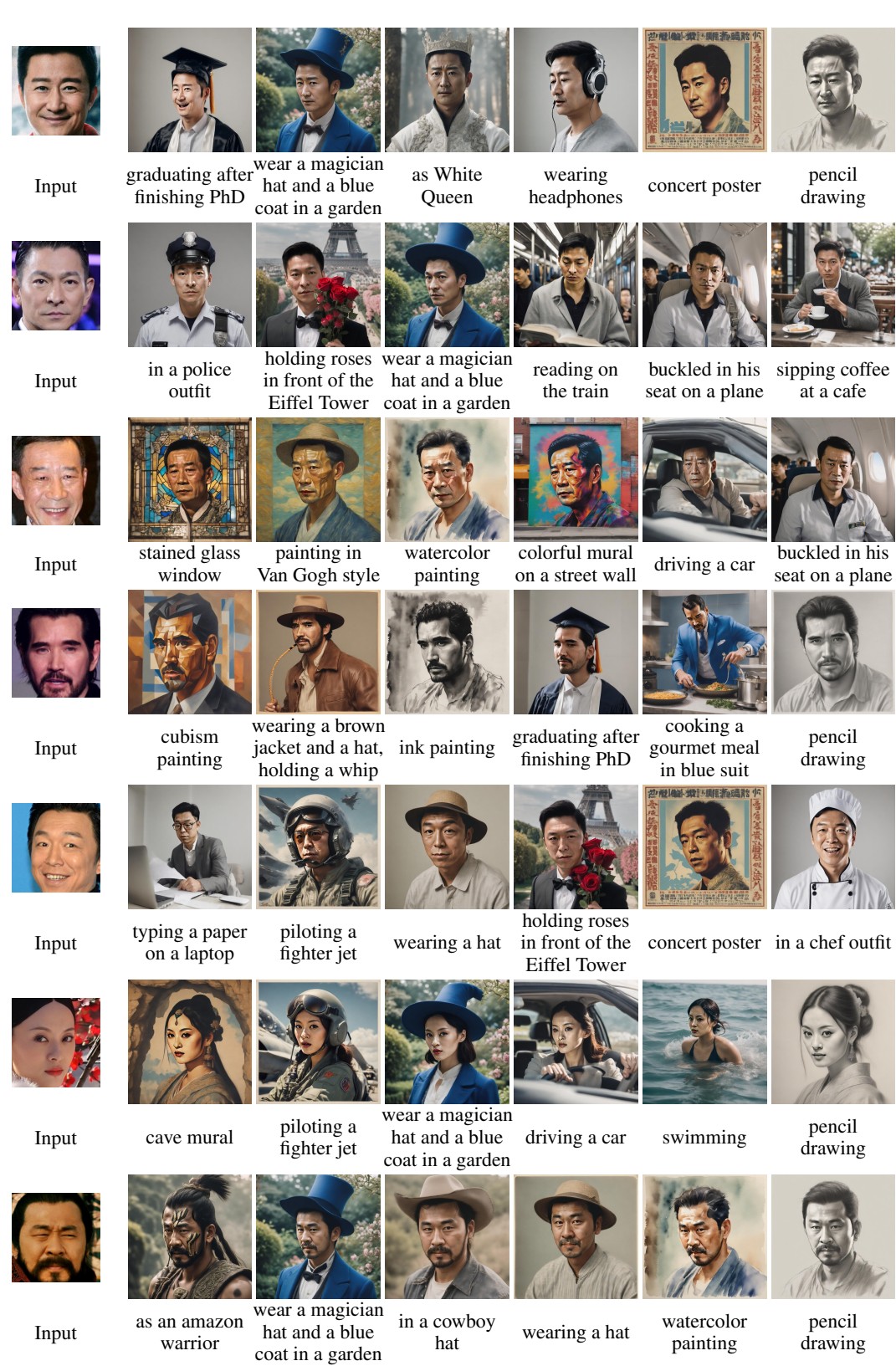

Figure 9: Additional face-centric personalization results with piecewise rectified flow (Yan et al., 2024) based on SDXL (Podell et al., 2024). Our method is compatible with more advanced base models and provides sophisticated personalization results.

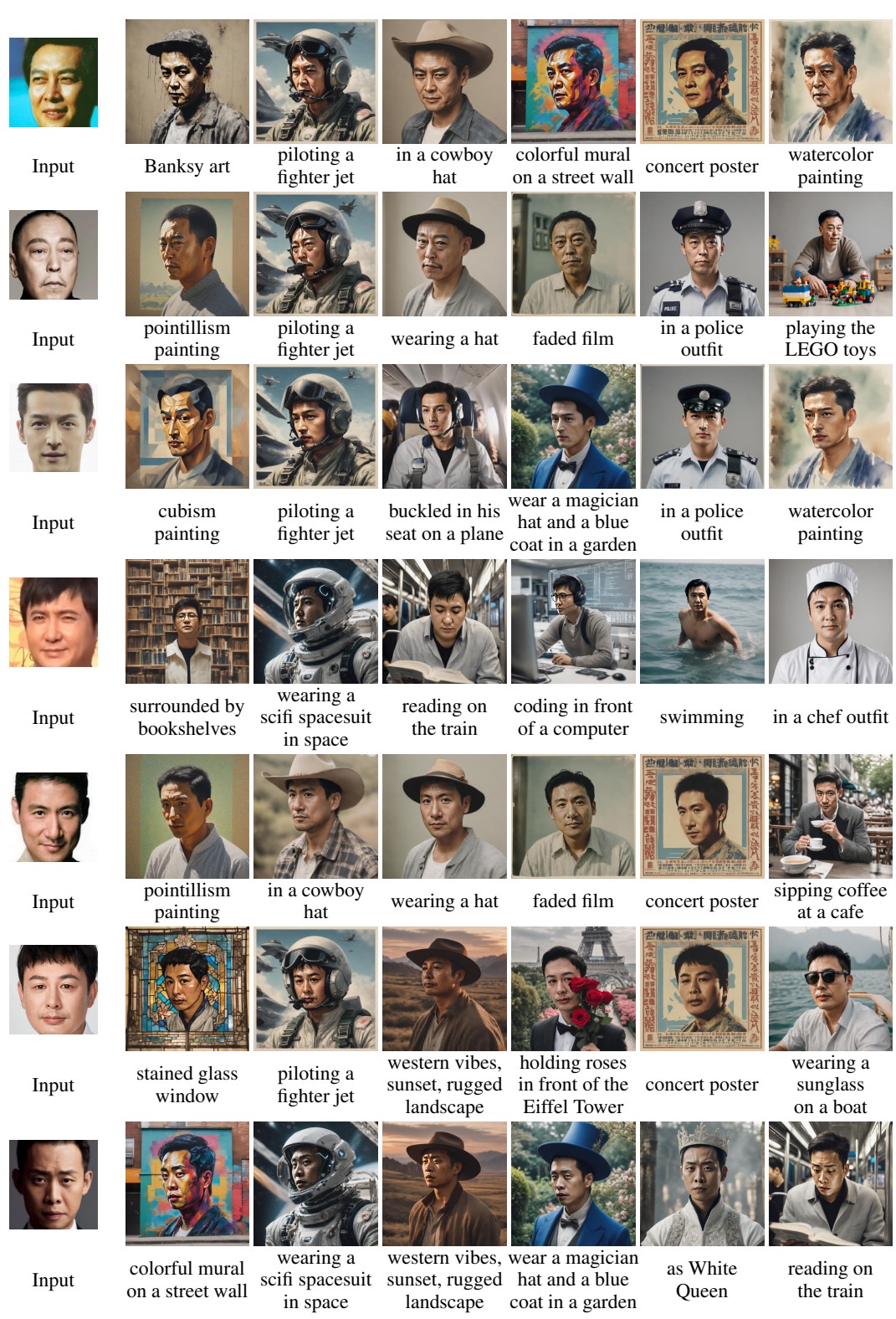

Figure 10: Additional face-centric personalization results with piecewise rectified flow (Yan et al., 2024) based on SDXL (Podell et al., 2024). Our method is compatible with more advanced base models and provides sophisticated personalization results.

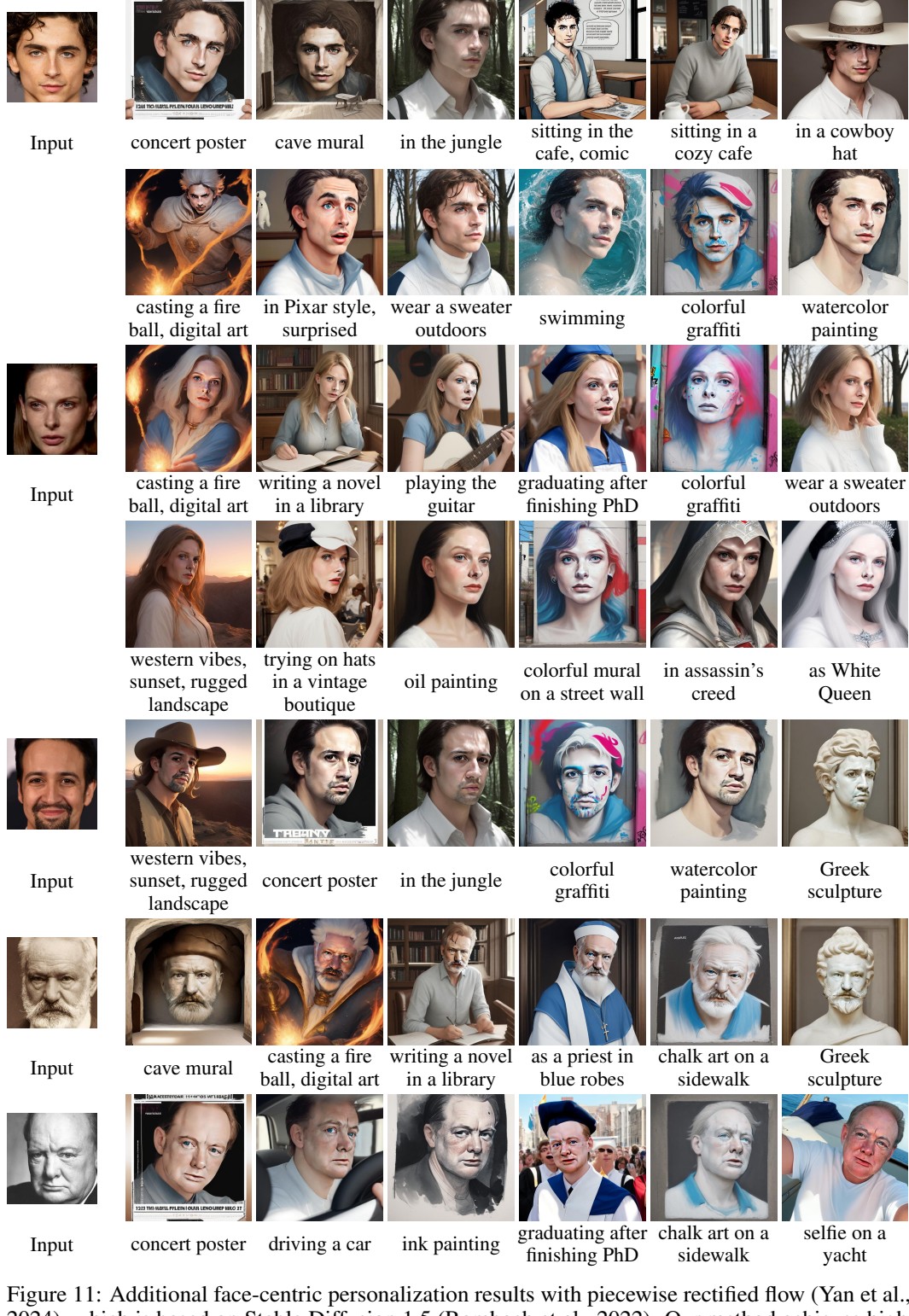

Figure 11: Additional face-centric personalization results with piecewise rectified flow (Yan et al., 2024), which is based on Stable Diffusion 1.5 (Rombach et al., 2022). Our method achieves high identity consistency. See Fig. 16 for results on the vanilla 2-rectified flow (Liu et al., 2023a, 2024b).

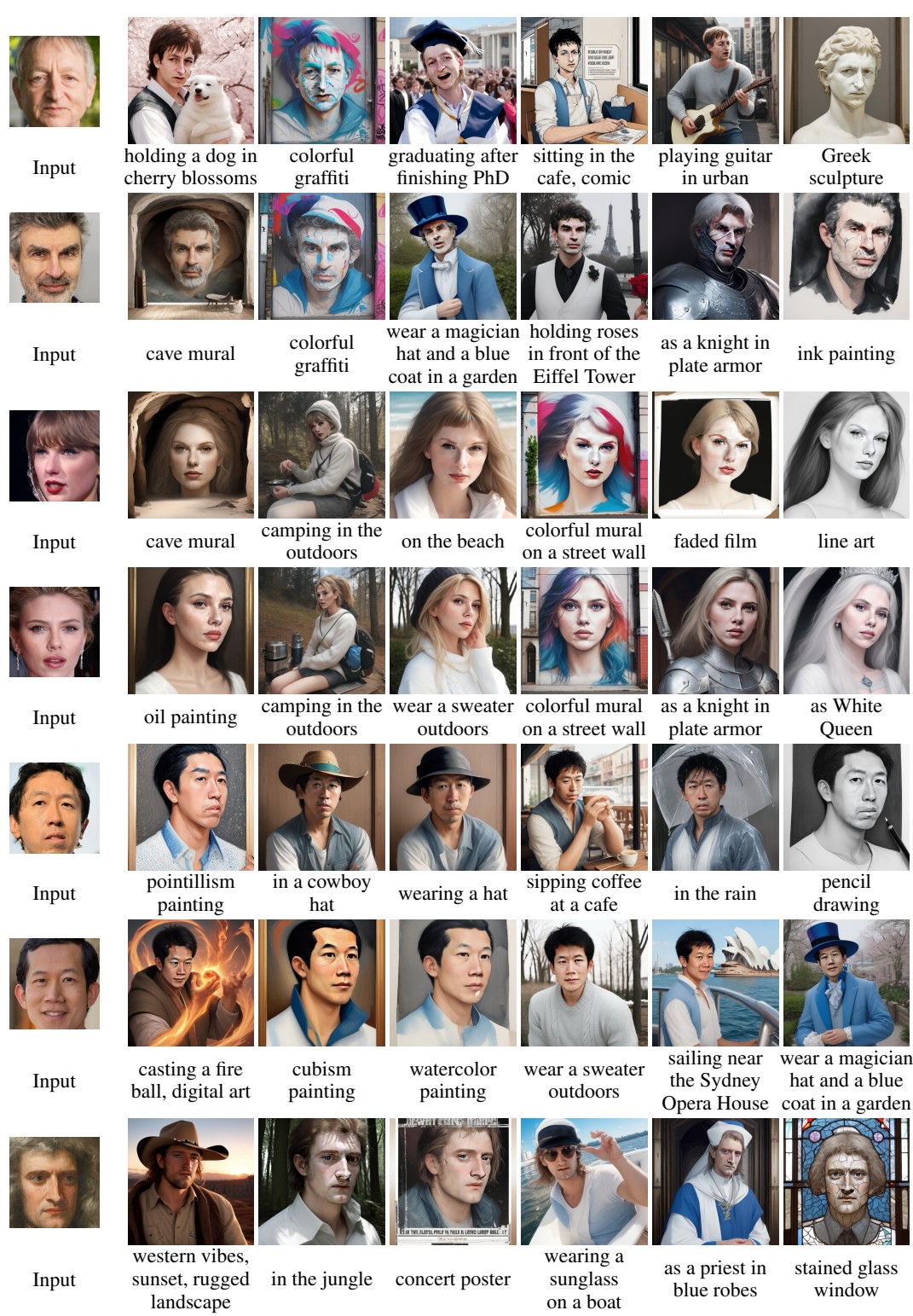

Figure 12: Additional face-centric personalization results with piecewise rectified flow (Yan et al., 2024), which is based on Stable Diffusion 1.5 (Rombach et al., 2022). Our method achieves high identity consistency. See Fig. 16 for results on the vanilla 2-rectified flow (Liu et al., 2023a, 2024b).

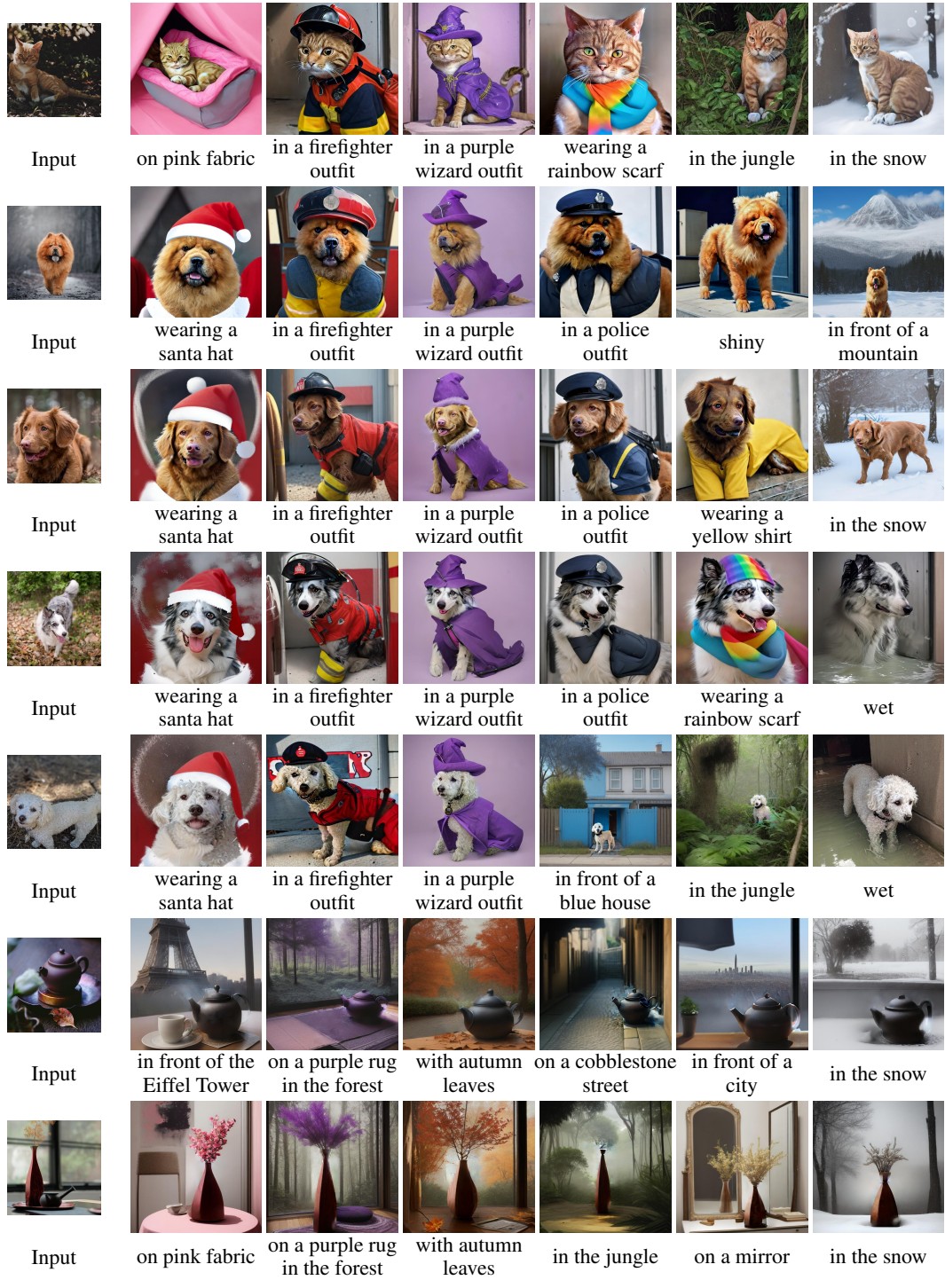

Figure 13: Additional subject-driven generation results with piecewise rectified flow (Yan et al., 2024), which is based on Stable Diffusion 1.5 (Rombach et al., 2022). Our approach preserves the identity of both live subjects and some regularly shaped objects. Please see Fig. 17 for more examples using the vanilla 2-rectified flow (Liu et al., 2023a, 2024b).

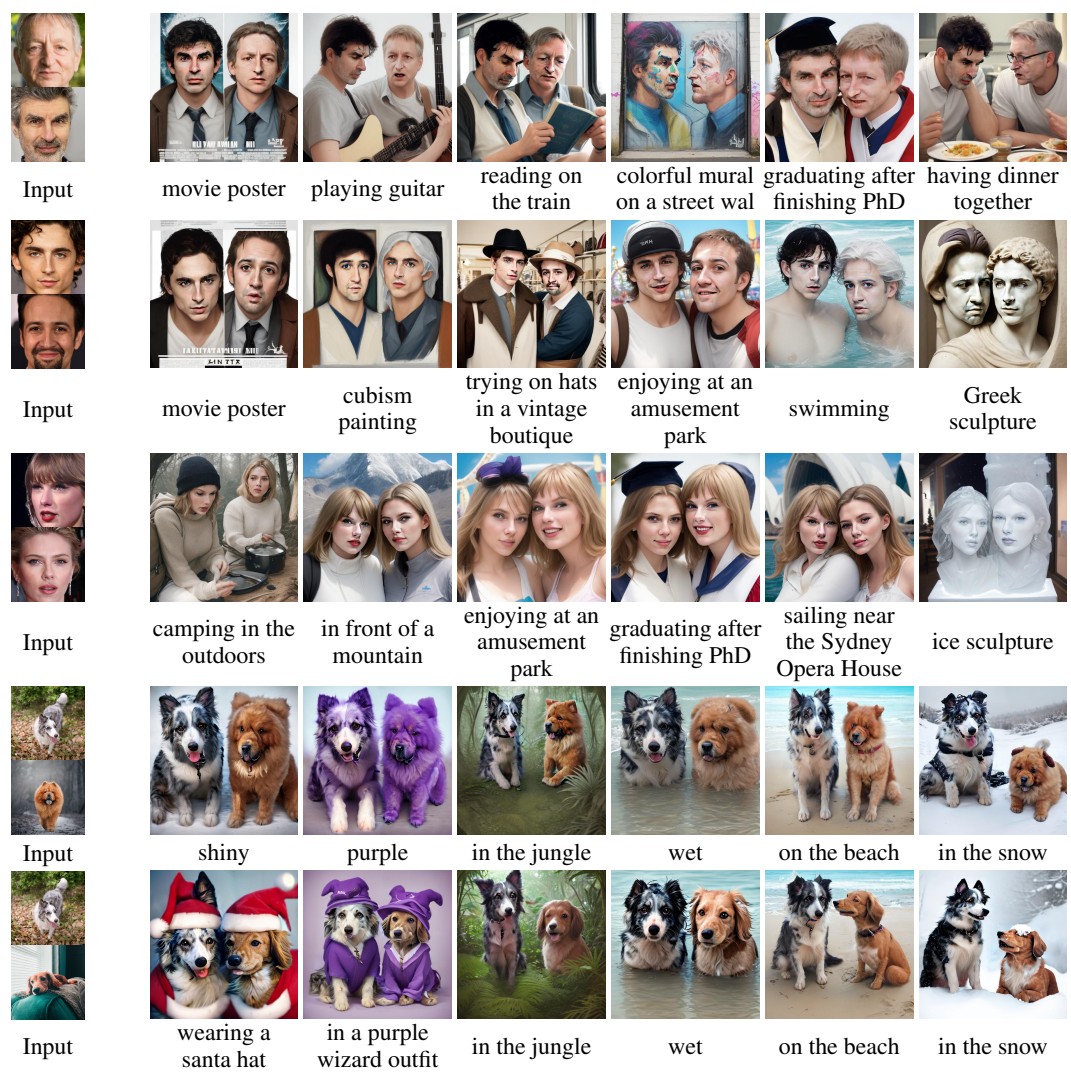

Figure 14: Additional multi-subject personalization results with piecewise rectified flow (Yan et al., 2024), which is based on Stable Diffusion 1.5 (Rombach et al., 2022). Our approach can naturally compose multiple subjects into the generated image while preserving their identities.

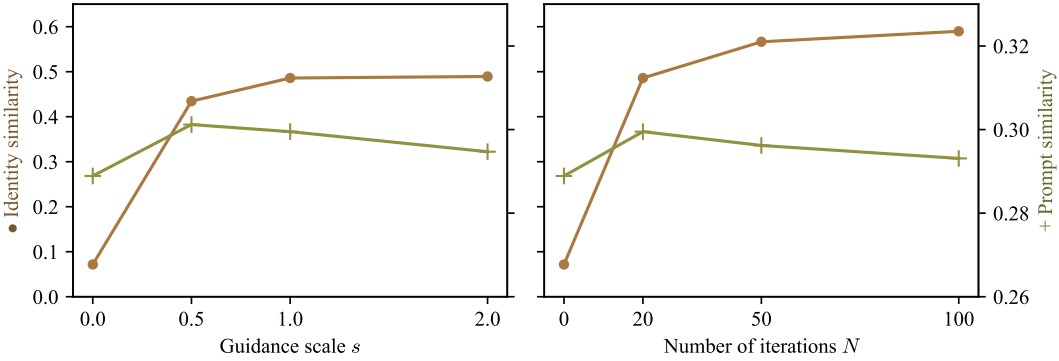

Figure 15: Ablation study of hyperparameters. Left: ablation study of guidance scale $s$ under $N = 20$. Right: ablation study of the number of iterations $N$ under $s = 1.0$. Our method remains effective over a reasonably wide range of hyperparameters.

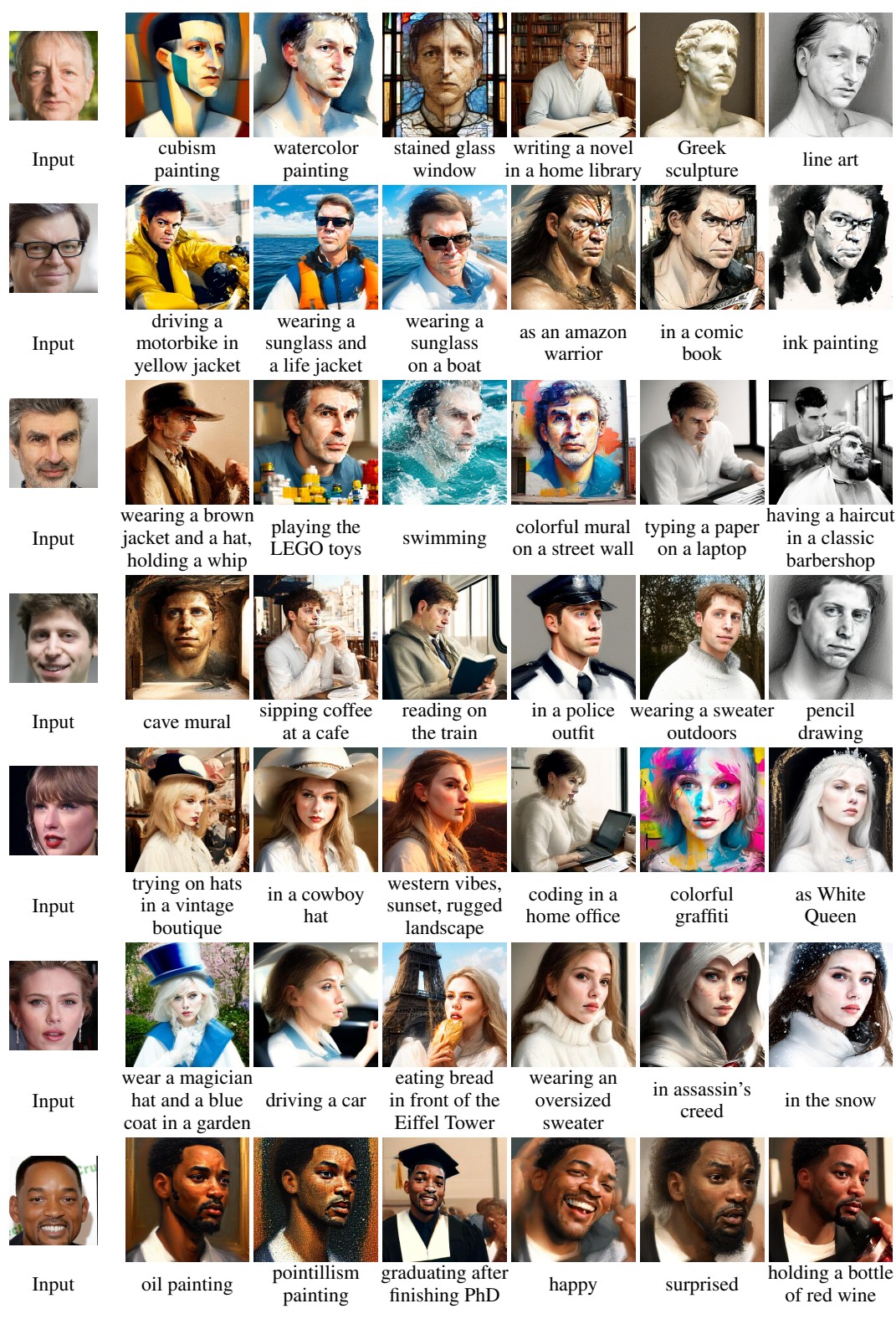

Figure 16: Face-centric personalization results with vanilla 2-rectified flow (Liu et al., 2023a, 2024b), which is based on Stable Diffusion 1.4 (Rombach et al., 2022). Our method preserves their identities well while remaining faithful to the text prompt during generation.

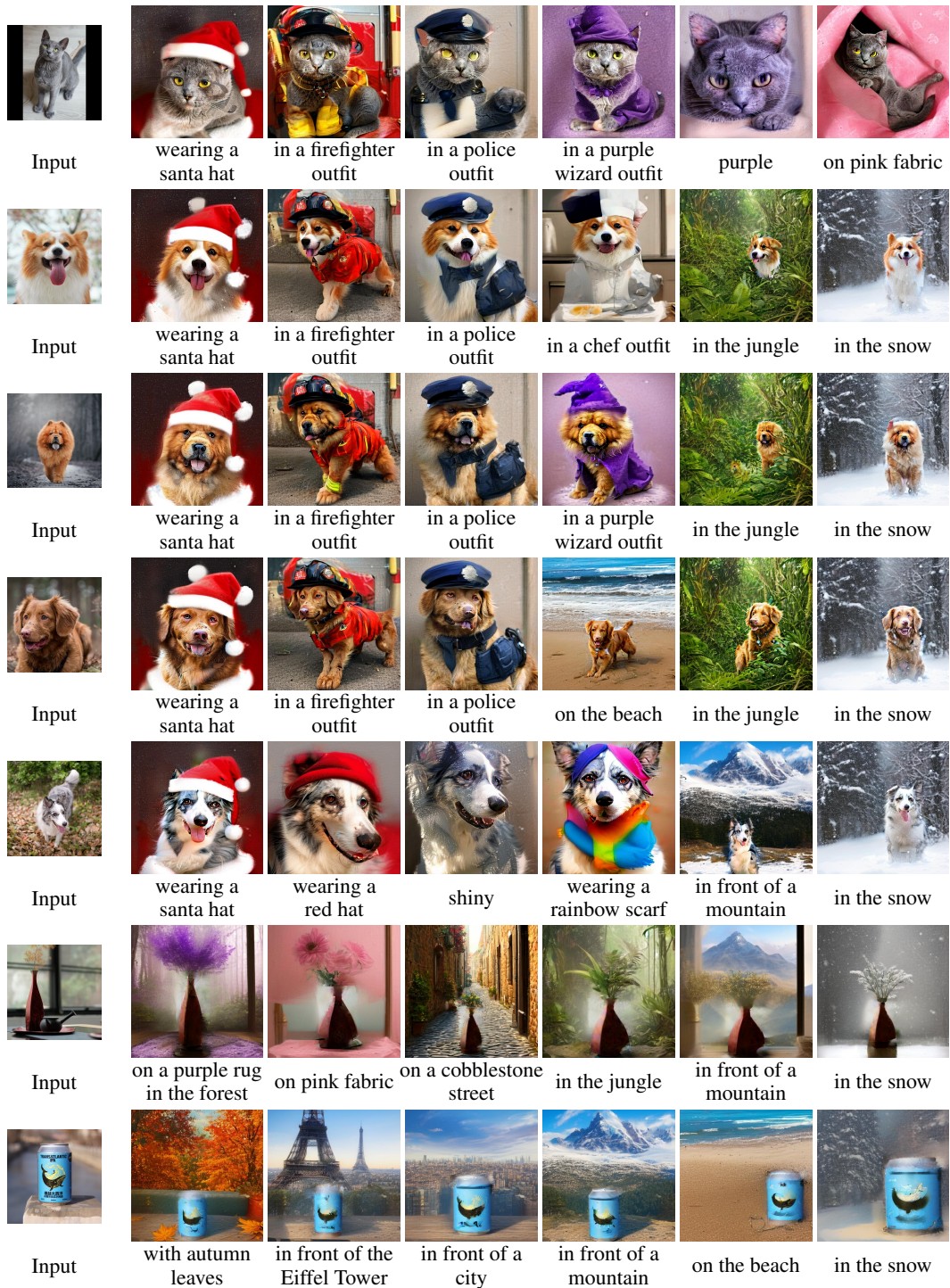

Figure 17: Subject-driven generation results with vanilla 2-rectified flow (Liu et al., 2023a, 2024b), which is based on Stable Diffusion 1.4 (Rombach et al., 2022). Examples for additional categories of cats, dogs, and objects are included to demonstrate the effectiveness of our approach.

