# OpenReview forum: "RectifID: Personalizing Rectified Flow with Anchored Classifier Guidance"
_NeurIPS.cc/2024/Conference — NeurIPS 2024 poster_

### Official Review · Reviewer_Fu9U · 2024-07-08

**Soundness:** 3
**Presentation:** 2
**Contribution:** 3
**Rating:** 6
**Confidence:** 4

**Summary:**

This paper introduces a new method for Rectified Flow models to perform classifier-guidance sampling without needing a noise-aware classifier. Specifically, the authors leverage a fixed-point method to overcome the need for a noise-aware classifier and anchor the classifier-guided flow trajectory to a reference trajectory to stabilize the sampling process. In practice, the authors apply the method to personalization tasks, transferring the identity of a face or an object from a reference image to new generations paired with customized prompts. The process looks promising as demonstrated in the paper.

**Strengths:**

The idea is simple and makes sense; the application task is also good and worthy.

**Weaknesses:**

Not as I can think of

**Questions:**

Not as I can think of

**Limitations:**

The idea only applies to rectified flow models right now, but I think it could be generalized to border diffusion models, it would be interesting to elaborate more on this in the future.

---

> ### Author Rebuttal · Authors · 2024-08-06
>
> Thank you for the very constructive comments. Below are our responses to the raised concern.
>
> [L1] Generalization to broader diffusion models
>
> * While our classifier guidance is derived based on rectified flow, the same idea can be generalized to some few-step diffusion models by assuming straightness of their trajectories within each time step. We empirically demonstrate this in Figure 2 in the global response with two popular diffusion models, SD-Turbo [1] and phased consistency model [2]. As the results indicate, our method effectively personalizes these diffusion models to generate identity-preserving images. We will continue to explore this approach for other generative models in future research.
>
> ---
>
> [1] Sauer, Axel, et al. Adversarial diffusion distillation. arXiv 2023.
>
> [2] Wang, Fu-Yun, et al. Phased consistency model. arXiv 2024.

---

> ### Comment · Reviewer_Fu9U · 2024-08-09
> **after rebuttal**
>
> sorry for the initial short review and thanks for the author's rebuttal. After reading other reviewer's comments and the author's rebuttal, I don't think I have more insights to add here. And I would like to keep my rating.

---

### Official Review · Reviewer_bvm9 · 2024-07-15

**Soundness:** 3
**Presentation:** 3
**Contribution:** 3
**Rating:** 5
**Confidence:** 4

**Summary:**

The paper introduces a training-free method based on rectified flow and classifier guidance for personalized image generation. The experimental results show that the proposed method performs better than other state-of-the-art baselines in generating personalized images for human faces, live subjects, and certain objects.

**Strengths:**

- The paper is well-written and easy to follow, and the motivation of the proposal is clear.
- The theoretical background supports the motivation and the proposal well.
- The experimental results present a significant improvement compared to recent existing works in the field.

**Weaknesses:**

- The theoretical justification or at least an intuition for Equation (6) is essential.
- Does the "Time" column in Table 1 refer to the training time or the inference time? Does calling the gradient with respect to a classifier affect the inference speed of the proposal compared to other baselines?

**Questions:**

- The theoretical justification or at least an intuition for Equation (6) is essential.
- Does the "Time" column in Table 1 refer to the training time or the inference time? Does calling the gradient with respect to a classifier affect the inference speed of the proposal compared to other baselines?

---

> ### Author Rebuttal · Authors · 2024-08-06
>
> We deeply appreciate your valuable suggestions, and we would like to address your main concerns as follows:
>
> [W1/Q1] Theoretical justification for Equation (6)
>
> * The intuition for Equation (6) is to shift the velocity toward data regions with higher class likelihood. Formally, we verify this using the ODE formulation in EDM [1], which states that:
>   $$
>   v(z_t,t)=-\\dot\\sigma(t)\\sigma(t)\\nabla_{z_t}\\log p(z_t),
>   $$
>    where $\\sigma(t)$ is a noise schedule. By Bayes' theorem, the desired class-conditional distribution satisfies:
>   $$
>   \\nabla_{z_t}\\log p(z_t|c)=\\nabla_{z_t}\\log p(z_t)+\\nabla_{z_t}\\log p(c|z_t).
>   $$
>   It turns out that the new velocity $\\hat v(z_t,t)$ in Equation (6) generates the desired distribution when the guidance scale is set to $s=-\\dot\\sigma(t)\\sigma(t)$:
>   $$
>   \\begin{aligned}
>   \\hat v(z_t,t)&=-\\dot\\sigma(t)\\sigma(t)\\nabla_{z_t}\\log p(z_t)+s\\nabla_{z_t}\\log p(c|z_t)\\\\
>   &=-\\dot\\sigma(t)\\sigma(t)\\nabla_{z_t}\\log p(z_t|c).
>   \\end{aligned}
>   $$
>   In practice, a different scale can be used to adjust the guidance, resulting in the form of Equation (6).
>
> [W2/Q2] Issues with the "Time" column in Table 1
>
> * The "Time" column in Table 1 refers to the inference time. Notably, our method is applied only at inference, without requiring additional training of the diffusion model. We will revise the table caption to make this clearer.
> * Regarding the inference time, there is indeed an overhead of about 0.2 seconds per iteration in calling the gradient w.r.t. the classifier. Nevertheless, the iterative process can quickly converge within 10 seconds over 20 iterations, which is comparable to other baselines in terms of efficiency . This efficiency is attributed to our proposed anchored classifier guidance, as confirmed by the ablation study, and we expect further speedup with algorithm improvements.
>
> ---
>
> [1] Karras, Tero, et al. Elucidating the design space of diffusion-based generative models. NeurIPS 2022.

---

> > ### Comment · Reviewer_bvm9 · 2024-08-12
> >
> > Thank you to the authors for their responses. I would like to keep my positive evaluation on the paper.

---

### Official Review · Reviewer_dDUa · 2024-07-17

**Soundness:** 3
**Presentation:** 3
**Contribution:** 3
**Rating:** 7
**Confidence:** 3

**Summary:**

The paper introduces a training-free method for subject-driven generation using diffusion models. This approach utilizes a new classifier guidance with off-the-shelf image discriminators and anchors the flow trajectory to a reference, ensuring stability and convergence. The method shows promising results in various personalization tasks for human faces, and other subjects.

**Strengths:**

1. The paper is well-written and easy to follow.

2. The methodology eliminates the need for extensive pre-training or subject-specific fintuning, making it highly eficient and adaptable to various use cases without the cost of training on large datasets​ or fintuning each model for each subject.

3. It allows for the use of off-the-shelf image discriminators, enabling not only personalization like presented in the paper but also other controllable generation tasks.

4. The idea of setting $t=1$ and solving Eq. 8 with fixed-point iteration is interesting.

**Weaknesses:**

1. The approach heavily relies on the availability and quality of pre-trained image discriminators, which may limit its applicability in domains lacking robust pre-trained models. For example, the method for personalizing live objects (dog, cat) is quite ad-hoc and engineering for me.

2. The scope of this paper is quite limited while the method is very general. Therefore, experiments on other tasks, as mentioned below in the Questions section, can further strengthen the paper.

**Questions:**

1. How to deal with the case of lacking pre-trained discriminators ?

2. The experimental settings are primarily focused on personalization with face-centric and specific object categories. However, the method is quite general. Have the authors tried other tasks like in Universal Guidance [1] ?

[1] Universal Guidance for Diffusion Models. Arpit Bansal, Hong-Min Chu, Avi Schwarzschild, Soumyadip Sengupta, Micah Goldblum, Jonas Geiping, Tom Goldstein. 2023

**Limitations:**

They have sufficiently addressed the limitations and potential societal impacts in their work.

---

> ### Author Rebuttal · Authors · 2024-08-06
>
> Thank you for providing valuable feedback. Here are our responses to the concerns raised.
>
> [W1/Q1] Domains lacking pre-trained discriminators
>
> * In the short term, we suggest first training a specialized discriminator and then applying our classifier guidance. There are two reasons for doing this instead of finetuning the generator directly: (1) training/finetuning a discriminator is usually more efficient and stable than training/finetuning a generator; (2) it can take full advantage of domain images that have no captions or even labels by using standard contrastive learning loss.
> * In the future, scaling up vision-language models may be a general solution for these domains. The current models such as GPT-4V have demonstrated certain generalizability across visual understanding tasks. As they continue to improve in generalizability and robustness, they will become a viable source for guiding diffusion models in new domains.
>
> [W2/Q2] More controllable generation tasks
>
> * Following your suggestion, we've extended our method to more controllable generation tasks by directly using the guidance functions from Universal Guidance [1]. The experimental results under the guidance of segmentation map or style image are illustrated in Figure 1 in the global response. As shown, our classifier guidance can perform both tasks without additional tuning. This confirms the adaptability of our approach for various controllable generation tasks.
>
> ---
>
> [1] Bansal, Arpit, et al. Universal guidance for diffusion models. ICLR 2024.

---

> > ### Comment · Reviewer_dDUa · 2024-08-11
> > **Reply by Reviewer**
> >
> > Thank authors for your additional experiments for controllable generation tasks and an interesting answer of using LLM for Q1. Here, the additional experiments demonstrate that their method can be applied to other tasks with good results so I raise my score to 7

---

### Author Rebuttal · Authors · 2024-08-06

We thank all reviewers for the insightful comments, which are important for improving our work. In response, we have meticulously prepared a PDF file containing figures that effectively address some of the raised concerns. Below is a concise summary of these figures.

* Figure 1: Results for more controllable generation tasks (Reviewer dDUa).
* Figure 2: Generalization to broader diffusion models (Reviewer Fu9U).

---

### Decision · Program_Chairs · 2024-09-25

**Decision:**

Accept (poster)

**Comment:**

This paper was reviewed by three experts in the field. The reviewers agreed on paper acceptance due to its clear writing, well-motivated problem, interesting solution, highly efficiency and adaptability, flexibility to extend for other tasks, and strong experimental results. The scores are Accept, Weak Accept, and Borderline Accept after the rebuttal. Based on the reviewers' feedback, the decision is to recommend the paper for acceptance to NeurIPS 2024.